# Cingulate-motor circuits update rule representations for sequential choice decisions

Daigo Takeuchi[1,2,3], Dheeraj Roy ®[1,4], Shruti Muralidhar[1,2], Takashi Kawai[1,2], Andrea Bari[1,2], Chanel Lovett[3], Heather A. Sullivan[5], Ian R. Wickersham ®[5] & Susumu Tonegawa ®[1,2,3,6] ✉

Anterior cingulate cortex mediates the flexible updating of an animal's choice responses upon rule changes in the environment. However, how anterior cingulate cortex entrains motor cortex to reorganize rule representations and generate required motor outputs remains unclear. Here, we demonstrate that chemogenetic silencing of the terminal projections of cingulate cortical neurons in secondary motor cortex in the rat disrupts choice performance in trials immediately following rule switches, suggesting that these inputs are necessary to update rule representations for choice decisions stored in the motor cortex. Indeed, the silencing of cingulate cortex decreases rule selectivity of secondary motor cortical neurons. Furthermore, optogenetic silencing of cingulate cortical neurons that is temporally targeted to error trials immediately after rule switches exacerbates errors in the following trials. These results suggest that cingulate cortex monitors behavioral errors and updates rule representations in motor cortex, revealing a critical role for cingulate-motor circuits in adaptive choice behaviors.

A central feature of animal intelligence is the hierarchical organization of behaviors and the capacity to adopt flexible strategies that allow complex sequential actions[1–3]. Previous studies in primates suggested that premotor cortex and supplementary motor areas underlie the planning and execution of sequential movements[4–7]. More recently, it has been demonstrated that neurons in the rodent secondary motor cortex (M2) similarly code initiation of sequential movements, motor planning, memory, and values for upcoming choice actions, suggesting that they maintain representations of sensorimotor associations for adaptive choice behavior[8–14]. These findings raise a question of what circuit mechanisms enable M2 neurons to process context-dependent information such as task rules. More specifically, how is this information provided from input brain regions and how are they processed by M2 circuits?

Studies in humans and other animals have shown that medial prefrontal areas including anterior cingulate cortex (ACC) mediate flexible decisions in the face of rule changes (e.g., task switching) or under uncertain conditions[15–20]. However, despite abundant anatomical evidence of cingulate projections to motor cortices[21–24], there are several fundamental issues that remain poorly understood. (1) Dimensions of task rule representations in ACC: what aspects of task rules ACC circuits represent? We addressed this issue by using a behavioral task in which rats were required to adapt their sequential choice behaviors between single-step and two-step choice responses. The rule switches in this task paradigm are asymmetric in their design (i.e., task switches between single-step and multi-step responses) and this feature allowed us to distinguish whether ACC circuits are recruited in rule switches that demand an increment of response steps (i.e., 1 step to 2 steps), a decrement of response steps (i.e., 2 steps to 1 step) or in both types of rule switches. (2) Relevance of ACC to motor output pathways: how does ACC entrain motor cortex to update neural representations for rules upon sudden task rule changes? To address this issue, we

[1]RIKEN-MIT Laboratory for Neural Circuit Genetics at the Picower Institute for Learning and Memory, Cambridge, MA, USA. [2]Department of Brain and Cognitive Sciences, MIT, Cambridge, MA, USA. [3]Howard Hughes Medical Institute, MIT, Cambridge, MA, USA. [4]Stanley Center for Psychiatric Research, Broad Institute of MIT and Harvard, Cambridge, MA, USA. [5]McGovern Institute for Brain Research at MIT, Cambridge, MA, USA. [6]Department of Biology, MIT, Cambridge, MA, USA. ✉e-mail: tonegawa@mit.edu

perturbed the projections from ACC to M2 using chemogenetic and optogenetic methods and examined what aspects of choice behaviors were mediated by the ACC→M2 circuit and how ACC circuits modulate neural activity in M2 after rule switches in which mapping of choice actions to rewards are suddenly changed. Combining these pathway-specific and task period-specific manipulations with animals' behaviors in a choice decision task and with neural activity measurements from M2, this study revealed a specific functional role of ACC→M2 pathway in flexible updating of rule representations upon rule switches during animals' sequential choice responses.

## Results

### Conditional action sequencing task

We devised a behavioral task in which an animal updates its sequential choice response for rewards upon abrupt rule changes (hereafter referred to as the conditional action sequencing task or CAS task). Briefly, in the CAS task, animals are required to choose left or right ports based on an auditory tone cue stimulus. Under the 1 step rule condition, animals received a reward after correctly poking a left or right port instructed by one of two tone cues, and could start the next trial after an inter-trial interval (ITI) (Fig. 1, top schematic). When animals nose-poked into an incorrect side port, they received no reward, and instead, a buzzer tone was delivered. Under the 2 steps rule condition, animals received a first reward after making a correct first choice and then received another reward after poking the opposite side port (Fig. 1, bottom schematic). If the animal pushed the center lever before choosing the opposite side port, they received no reward and instead heard the buzzer tone. Rules were switched between these two conditions every 55 trials, requiring the animals to adjust their choice behavior driven by water rewards or buzzer penalties (Supplementary Fig. 1). We trained rats to achieve over 70% success rate for both 1 step and 2 steps conditions, and for both tone cues.

### Silencing anterior cingulate excitatory neurons disrupted the animal's ability to adapt to rule switches

To examine the potential roles of ACC in animals' flexibly adapting to rule switch in the CAS task, we chemogenetically silenced neural activity in ACC and examined its effect on animals' task performance. First, we bilaterally injected an inhibitory DREADD virus in the ACC of rats that had been trained on the CAS task (Fig. 2a and Supplementary Fig. 2a). Intraperitoneal (IP) administration of clozapine-N-oxide (CNO) decreased spiking activity in ACC, an effect that lasted over 60 min after reaching the plateau level of firing (Supplementary Fig. 2b). After animals recovered from virus injection surgeries, we injected CNO or saline and tested their 2 steps choice performance for one block followed by another block after a rule switch. To quantify the animal's ability to adapt to rule switches from 1 step to 2 steps rules, we measured the animal's second choice performance (%2nd choice omission error) (see Supplementary Fig. 1d for details). Similarly, to quantify the animal's ability to adapt to rule switches from 2 steps to 1 step rules, we measured the frequency of 2nd choice commission error (Supplementary Fig. 1d). In a representative saline session (Fig. 2b, c and

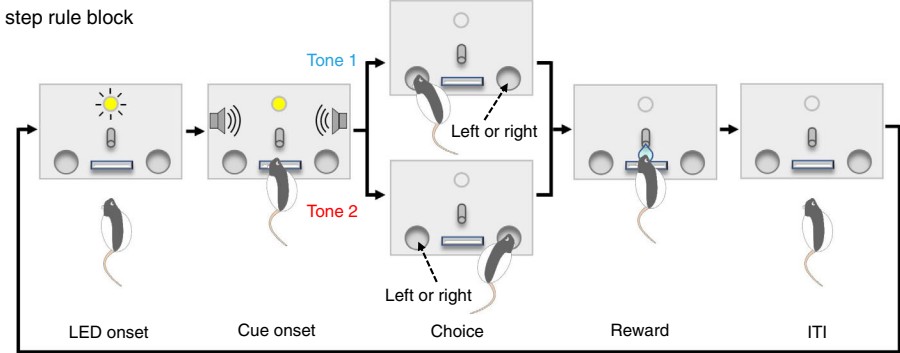

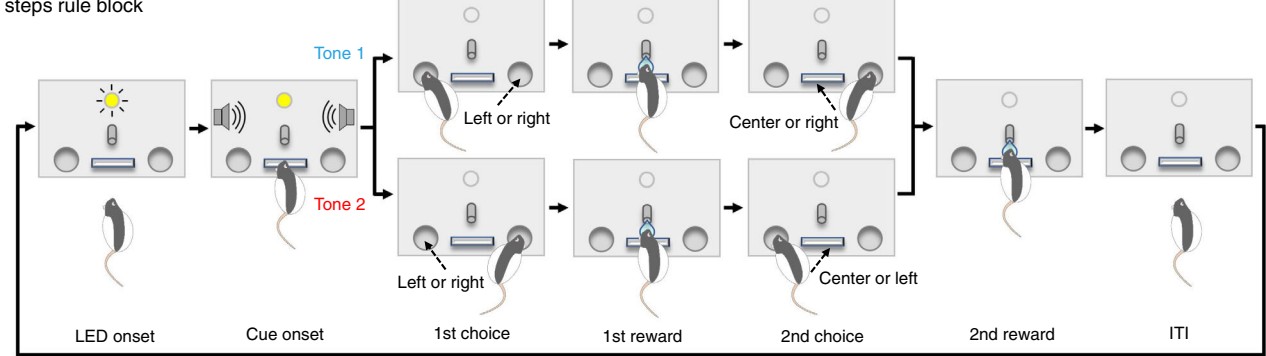

**Fig. 1 | Conditional action sequencing task.** Rats were trained and tested in a chamber in which a lever, a water spout, and an LED were installed on the front wall with two infrared (IR) ports on the left and right sides. Sound speakers were equipped on side walls. Task rules were switched between 1 step rule and 2 steps conditions every 55 trials. LED onset signals the end of the inter-trial interval (ITI) and rats can start a new trial by pushing the center lever. In 1 step condition (top), animals received a water reward after correctly poking the left or right IR port as instructed by one of two tone cues. When animals poked an incorrect port, they received no reward, and instead, a buzzer sound was delivered with an extra waiting being imposed in ITI before starting the next trial (i.e., 1st choice error; also see Supplementary Fig. 1c). In 1 step condition, no reward but an error feedback buzzer followed by an extra waiting time was delivered when animals made a second choice to the opposite side port (2nd choice commission error; also see Supplementary Fig. 1d). In 2 steps condition (bottom), animals received a reward after making a correct first choice and then received another reward after poking the opposite side port. If an animal made an incorrect first choice, no reward was delivered. Instead, a buzzer sound was delivered, and an extra waiting time was imposed in ITI before starting the next trial (1st choice error). If the animal made a correct first choice but pushed the center lever before poking the opposite side port, it received no reward. Instead, a buzzer sound was delivered, and an extra waiting time was imposed in ITI (2nd choice omission error; also see Supplementary Fig. 1d).

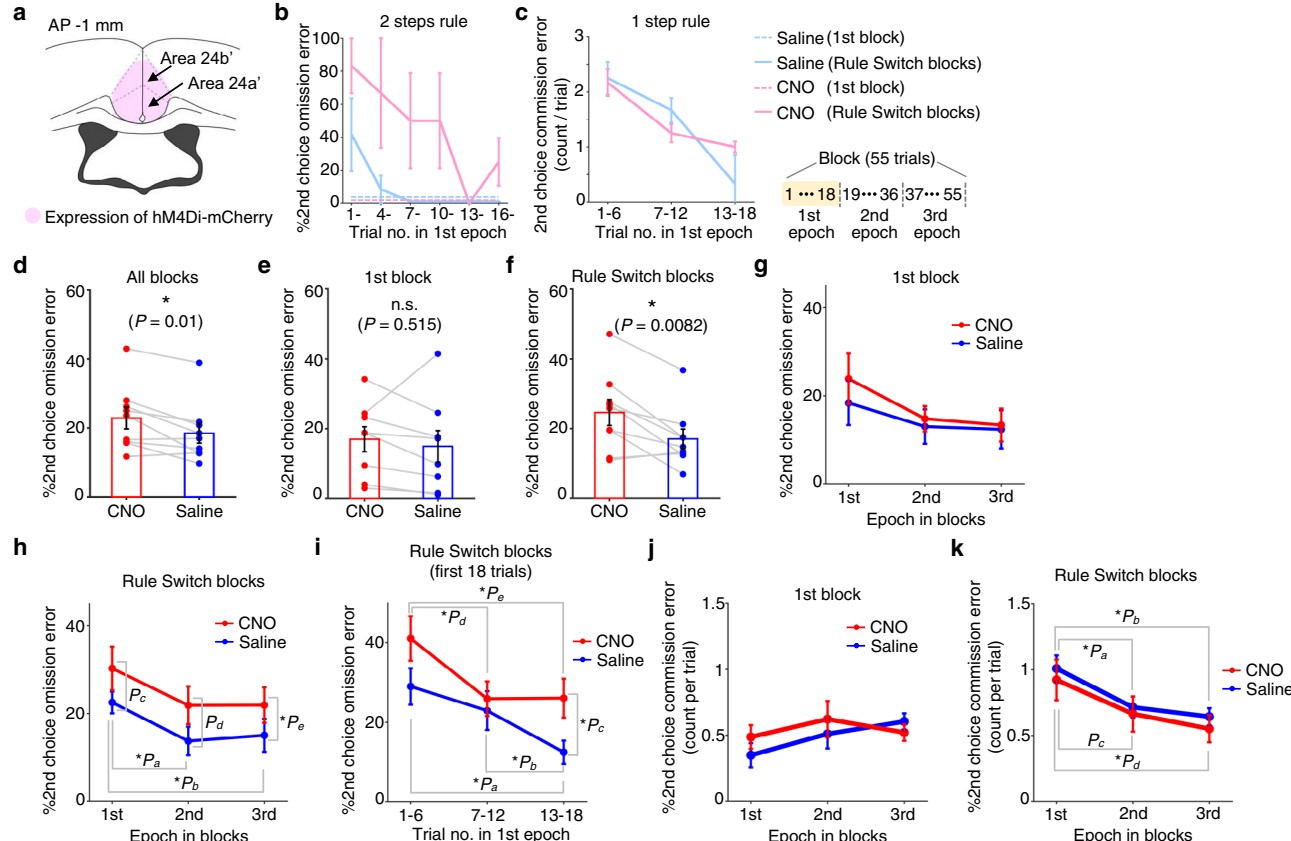

**Fig. 2 | Chemogenetic silencing of ACC affected task performance. a** Inhibitory DREADD virus was injected in ACC. **b** 2nd choice performance for 1st epoch in 2 steps condition in representative CNO and saline sessions (three rule switches each for 1→2 steps and for opposite direction in both sessions). Blue, saline. Pink, CNO. Dotted and solid lines, 1st and Rule Switch-blocks. **c** 2nd choice commission error for 1st epoch in 1 step condition for the same sessions. **d** 2nd choice performance for all 2 steps blocks. Blue, saline. Red, CNO. **e** Same as in **d**, but for 1st block. **f** Same as in **d**, but for Rule Switch-blocks. **g** 2nd choice performance for 1st block in 2 steps condition. Repeated measures ANOVA revealed no main effect ($P > 0.4$ for dose; $P > 0.1$ for epoch). **h** Same as in **g**, but for Rule Switch-blocks. CNO dose showed a main effect ($P = 0.018$, $F_{1,50} = 5.98$ for dose; $P = 0.051$, $F_{2,50} = 3.16$ for epoch) with no interaction ($P > 0.9$). Post hoc comparisons using two-sided paired

$t$-test with Bonferroni's correction across epochs (no correction for dose because it contained only two conditions): $*P_a = 0.009$; $*P_b = 0.015$; $P_c = 0.11$; $P_d = 0.054$; $*P_e = 0.021$. **i** 2nd choice performance for 1st epoch of Rule Switch-blocks in 2 steps condition. $*P_a = 0.0342$; $*P_b = 0.0050$; $*P_c = 0.044$; $*P_d = 0.025$; $*P_e = 0.0047$. **j** 2nd choice commission error for Rule Switch-blocks in 1 step condition. ANOVA revealed no main effect ($P > 0.6$ for dose; $P > 0.4$ for epoch). **k** Same as in **j**, but for Rule Switch-blocks. Significant main effect for epoch ($P = 0.0038$, $F_{2,50} = 6.26$) but not for dose ($P > 0.3$) with post hoc comparisons: $*P_a = 0.00042$, $*P_b = 0.00016$, $P_c = 0.038$, and $*P_d = 0.013$. $n = 3$ Rule Switch-blocks each for CNO and saline sessions (**b, c**). All pairwise comparisons were conducted using a two-sided paired $t$-test with $n = 9$ (**d–f, h–k**) or 8 rats (**g**). Error bar, SEM (**b–k**). Source data are provided as a Source Data file.

Supplementary Figs. 3 and 4), second choice errors (2nd choice omission error) were observed only in the first few trials after rule switches, indicating that the animal could adapt to rule switches from 1 step to 2 steps rules within a few trials (Fig. 2b, thick blue line. Also see right panels in Supplementary Fig. 3a, b). In contrast, when the same animal received CNO in another session, second choice errors persisted beyond trials that immediately followed rule switches from 1 step to 2 steps rules (Fig. 2b, thick pink line. Also see left panels in Supplementary Fig. 3a, b), suggesting that silencing ACC neurons impaired the animal's ability to adapt to rule switches from 1 step to 2 steps rules. Similarly, the animal showed 2nd choice commission error more frequently in trials that immediately followed rule switches from 2 steps to 1 step, which decreased within 10–20 trials after rule switches (Fig. 2c, blue), indicating that the animal could adapt to rule switches from 2 steps to 1 step conditions (Fig. 2c and Supplementary Fig. 3a). The same tendency was observed in CNO session (Fig. 2c), suggesting that silencing ACC neurons did not impact the animal's ability to adapt to rule switches from 2 steps to 1 step rules. We repeated these experiments using eleven rats injected with inhibitory DREADDs in ACC. Results showed that the 2nd choice error rate (%2nd choice omission error) in the 2 steps condition was significantly greater for the CNO condition (Fig. 2d. Also see Supplementary Fig. 5a

for individual animals' data). In contrast, no difference was observed in the 1st choice performance in the 2 steps condition or in that of the 1 step condition (Supplementary Fig. 5b, c), indicating that chemogenetic silencing of ACC excitatory neurons did not affect the animals' ability to discriminate between the two auditory cues or their ability to make choice responses. To examine if chemogenetic silencing of ACC neurons impaired the animals' ability to make sequential actions or the ability to adjust their choice behavior according to rule changes, we calculated the 2nd choice performance (%2nd choice omission error) in 2 steps rule for the 1st block of the session (i.e., non-rule switching block) and for subsequent Rule Switch-blocks separately. We found a significant difference in the error rate between CNO and saline conditions for Rule Switch-blocks while there was no significant difference for the 1st block (Fig. 2e, f). This suggests that the silencing of ACC neural activity affected the animals' ability to adjust their responses to rule changes from 1 step to 2 steps more severely than their ability to make sequential actions. We next split the 2 steps block into three epochs and compared 2nd choice performance among these epochs. We found that, in the saline condition, animals committed 2nd choice errors more frequently in the 1st epoch (i.e., trials that immediately followed rule switches) than in the 2nd or 3rd epochs (Fig. 2h, blue). The difference in 2nd choice performance between saline and CNO

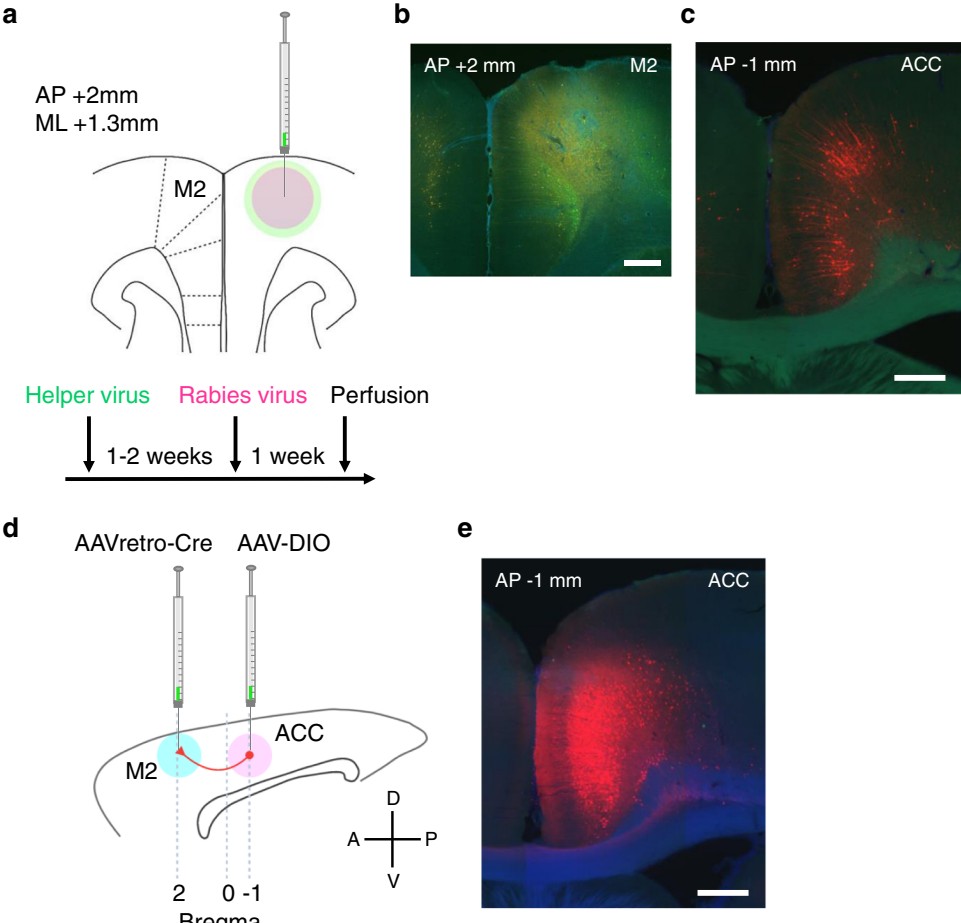

**Fig. 3 | Anatomical projections from ACC to M2. a** Anatomical projections from ACC to M2 were visualized using a genetically modified rabies virus system. One to two weeks after helper virus injection in M2 (a cocktail solution of AAV1-synP-FLEX-sTpEpB and pENN.AAV.CaMKII.0.4.Cre.SV40), rabies virus (RVΔG-4mCherry) was injected at the same coordinate. **b** A coronal section of the virus injection site in M2 (this is a magnified view of a region pointed by the red arrow in panel no. 4 in Supplementary Fig. 8a). Neurons infected by helper virus expressed GFP (coded in green in the image). Scale bar, 0.5 mm. **c** ACC neurons that were retrogradely infected with rabies virus expressed mCherry (coded in red in the image). Scale bar, 0.5 mm. **d** Projections from ACC to M2 were validated using AAVretro virus. Sagittal view of rat brain. Rats were injected with AAVretro-pmSyn1-EBFP-cre and AAV5-hSyn-DIO-hM4Di-mCherry viruses in M2 and ACC, respectively. **e** ACC neurons that were infected with AAVretro virus expressed mCherry (coded in red in the image) after Cre recombination. Scale bar, 0.5 mm.

conditions was smallest in the 1st epoch and increased toward the 3rd epoch (Fig. 2h, red and blue) and the high 2nd choice error rate (>20%) persisted beyond the 1st epoch in the CNO condition (Fig. 2h, red). These effects were observed in Rule Switch-blocks (Fig. 2h) but not in the 1st block (Fig. 2g), suggesting that silencing of ACC neural activity impaired the animal's ability to adjust choice responses upon rule switches from 1 step to 2 steps (see Supplementary Fig. 5f, g for individual animals' data). We further divided the 1st epoch (i.e., the first 18 trials immediately after rule switches) into three periods (1–6th, 7–12th, 13–18th trials) and compared the animals' 2nd choice performance between CNO and saline conditions (Fig. 2i). We found a significant difference in the 2nd choice performance between CNO and saline conditions in the 3rd period of the 1st epoch (i.e., corresponding to the 13–18th trials after rule switches), indicating that, on average, the impairment in adjusting choice responses to rule changes from 1 step to 2 steps due to the silencing of ACC neural activity showed up within the first 10–20 trials in the 2 steps rule block. mCherry control animals did not show any significant difference in the 2nd choice performance between CNO and saline conditions, excluding the possibility that the observed effect of chemogenetic silencing on task performance in previous experiments was caused by CNO administration itself (Supplementary Fig. 6).

We next compared the average number of 2nd choice commission error per trial across all three epochs in the 1 step condition. Results showed a significant effect of epoch in Rule Switch-blocks but not in the 1st block (Fig. 2j, k), indicating that animals could adapt to rule switches from 2 steps to 1 step rules (also see Supplementary Fig. 5d, e). Interestingly, no effect was found for CNO dose, suggesting that, while the silencing of neural activity in ACC impaired animals' ability to adapt to rule switches from 1 step to 2 steps rules, it did not affect the ability to adapt to rule switches in the opposite direction (i.e., from 2 steps to 1 step rules) (see Supplementary Fig. 5h, i for individual animals' data). Group analysis showed no difference in response latencies between saline and CNO conditions (Supplementary Fig. 7).

**Chemogenetic silencing ACC neuronal terminals in M2 disrupted the animals' ability to adapt to rule switches**

Given the abundant anatomical evidence of projections from ACC to motor cortices[21–24], we hypothesized that projections from ACC entrain motor cortex neurons to update neural representations for rules that are maintained in the motor circuit for generating required motor outputs. We first tried to identify anatomical projections from ACC to motor cortices using a genetically modified rabies virus approach (Fig. 3a)[25]. We found that a posterior portion of M2 (an area referred to

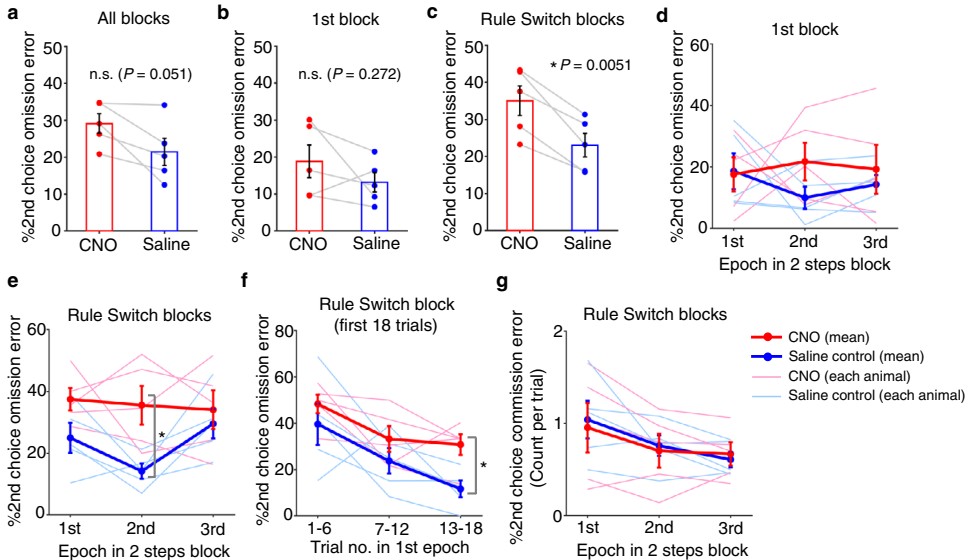

**Fig. 4 | Chemogenetic silencing of ACC neuronal terminals in M2 disrupted animals' choice performance. a** Group result of 2nd choice performance in 2 steps condition with a local infusion of either saline or CNO solution. **b** Same as in **a**, but % 2nd choice omission error for trials in 1st block (i.e., non-rule switching block). **c** Same as in **a** and **b**, but %2nd choice omission error for trials in Rule Switch-blocks. **d** 2nd choice performance was plotted separately for three epochs in the 1st block for sessions with a local infusion of saline or CNO solution. Thick red and blue lines represent across-animal averages of CNO and saline conditions, respectively ($n = 5$ rats). Thin lines represent individual animals. 1st, 2nd, and 3rd epochs correspond to 1–18th, 19–36th, and 37–55th trials. Neither CNO dose nor epoch in 2 steps rule block showed a main effect ($P > 0.2$ for CNO dose and $P > 0.9$ for epoch). **e** Same format as in **d**, but for Rule Switch-blocks. CNO dose showed a significant main effect ($P = 0.00402$, $F_{1,26} = 9,969$) while epoch did not ($P > 0.3$). Post hoc

comparisons were conducted using paired $t$-test (two-sided) with Bonferroni's correction across epochs (such correction was not conducted for dose because there are only two conditions, i.e., saline and CNO conditions). *$P = 0.0483$. **f** 2nd choice performance in 2 steps condition (%2nd choice omission error) was plotted separately for 1–6th, 7–12th, and 13–18th trials in the 1st epoch of Rule Switch-blocks. *$P = 0.0432$. **g** Average number of 2nd choice commission error per trial was plotted separately for three epochs in Rule Switch-blocks of 1 step condition. Repeated measures two-way ANOVA with both CNO dose and epoch being within-subject factors revealed no main effect of CNO ($P > 0.8$). Epoch showed a moderate effect but did not reach statistical significance ($P = 0.105$, $F_{2,26} = 2.461$, $n = 5$ rats). All pairwise comparisons were conducted using a two-sided paired $t$-test with $n = 5$ rats (**a**–**c**, **e**, **f**). Error bars, SEM (**a**–**g**). Source data are provided as a Source Data file.

as the frontal orienting field in previous studies and hereafter referred to as M2)[8,26] received projections from ACC (area 24a'/24b') (Fig. 3b, c and Supplementary Fig. 8)[27,28]. We also validated these projections from ACC to M2 using AAVretro-Cre and AAV-DIO viruses (Fig. 3d, e).

We then asked what aspects of choice behaviors are mediated by the ACC→M2 circuit that we identified. More specifically, we wanted to examine the potential roles of projections from ACC to M2 in the animal's ability to update sequential choices upon rule switches. Using bilateral infusion cannulae targeting M2 in rats that had been injected with the inhibitory DREADD virus in ACC, we infused CNO solution in M2 and examined its effect on task performance (Supplementary Fig. 9a, b). The 2nd choice omission error rate in 2 steps condition showed an increasing trend in the CNO condition as compared to saline control (Fig. 4a). In contrast, no difference was observed in the 1st choice performance in the 2 steps condition or in that of the 1 step condition (Supplementary Fig. 9c, d). We separately calculated 2nd choice performance in the 2 steps condition for the 1st block and for subsequent Rule Switch-blocks. The 2nd choice error rate was greater in the CNO condition relative to saline in Rule Switch-blocks but not for the 1st block (Fig. 4b, c), suggesting that chemogenetic silencing of ACC terminals in M2 impaired the animals' ability to adjust its choice responses to rule changes from 1 step to 2 steps more severely than its ability to make sequential actions. We next split the 2 steps rule blocks (55 trials) into three epochs (18, 18, and 19 trials for 1st, 2nd, and 3rd epochs, respectively) and compared the 2nd choice performance across epochs. Results showed a significant effect of CNO in Rule Switch-blocks while no such effect was found in the 1st block (Fig. 4d, e). We further divided the 1st epoch (i.e., the first 18 trials immediately after rule switches) in Rule Switch-blocks into three periods (6 trials each) and compared the animals' 2nd choice performance between

CNO and saline conditions (Fig. 4f). We found a significant difference in the 2nd choice performance between CNO and saline conditions in the 3rd period (i.e., corresponding to the 13–18th trials after rule switches), indicating that, like the IP injection experiments (Fig. 2i), the impairment in adjusting choice responses to rule changes showed up within the first 10–20 trials in the 2 steps rule block. We also compared the frequency of 2nd choice commission errors across all three epochs in the 1 step block. Like the results obtained in IP injection experiments (Fig. 2k), a decreasing tendency of 2nd choice commission error across epochs in Rule Switch-blocks of the 1 step condition was observed but no difference was found in the 2nd choice commission error frequency between CNO and saline conditions in either of the three epochs (Fig. 4g). These results suggested that the silencing of ACC terminals in M2 impaired the animals' ability to adjust their choice responses upon rule switches from 1 step to 2 steps rules but did not affect their ability to adapt to rule switches in the opposite direction (i.e., from 2 steps to 1 step rules) (also see Supplementary Fig. 9e). Finally, we conducted chemogenetic silencing of the prelimbic/infralimbic cortex and the ventral thalamic nuclei. We found no effect in any aspect of the task performance in both experiments (Supplementary Figs. 10 and 11 for results of prelimbic/infralimbic cortex and of ventral thalamic nuclei, respectively), indicating that ACC and its projections to M2, but not prelimbic/infralimbic cortex or ventral thalamic nuclei are specifically recruited for reorganizing sequential choice decisions upon rule switches.

### Chemogenetic suppression of ACC neural activity decreased rule selectivity in M2 neurons

We next asked how ACC circuits affect neural activity in M2. We unilaterally implanted array electrodes in M2 and measured spiking

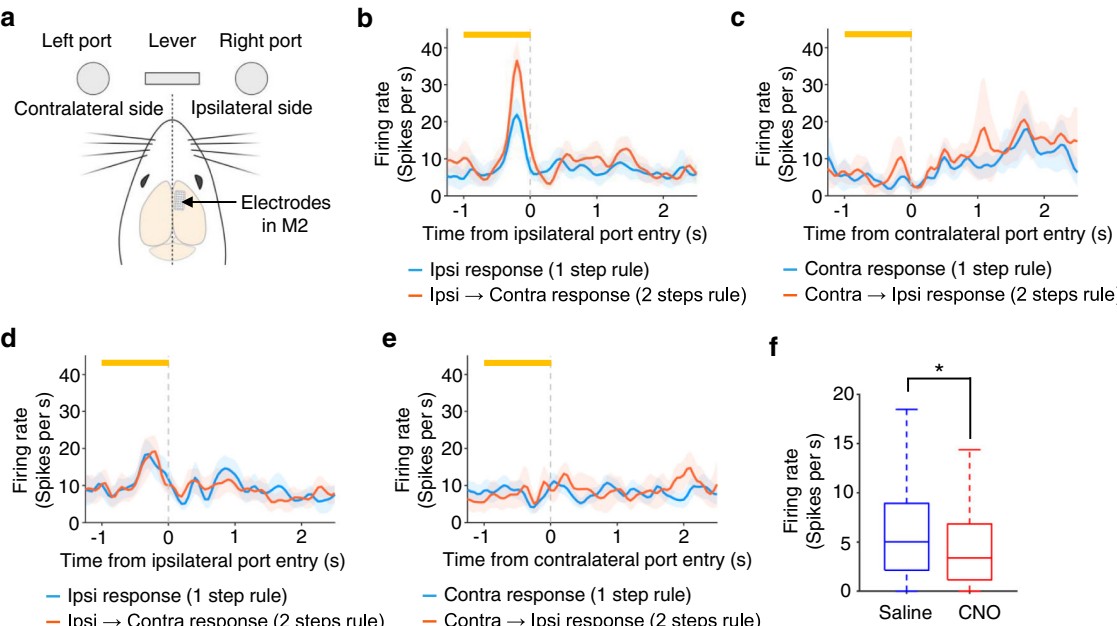

**Fig. 5 | Chemogenetic silencing of ACC decreased firing rate during pre-choice period in M2 neurons. a** Ipsilateral and contralateral sides viewed from a rat's cerebral hemisphere in which M2 single-unit activity was measured. Rats were implanted with Utah array electrodes in M2 of either left or right hemisphere and neural activity was measured during their task performance with IP injections of saline or CNO solutions (20 mg/kg). In the following analysis, trials were classified according to which side port (ipsilateral or contralateral) rats chose as their 1st choices. **b** Peri-event time histogram (PETH) of a representative single-unit showing rule selective responses before making a correct response to the side port that was located on the ipsilateral side of neural activity measurements (ipsilateral condition). In 1 step condition (blue line), the rat made a choice of the ipsilateral port while, in 2 steps condition (red line), the rat made its 1st choice of the ipsilateral port and then made its 2nd choice of the contralateral port. Only trials in Rule Switch-blocks were used for constructing the PETH. Orange bar at the top, a 1-s period immediately before rat's entry to the ipsilateral port (pre-choice period). The center of the band, mean. **c** Same as in **b**, but PETH was calculated using trials in which the rat made a correct response to the side port that was located on the contralateral side of neural activity measurements (contralateral condition). **d** PETH of a single-unit measured in a session in which the rat received an IP injection of CNO solution. **e** Same as in **d**, but for contralateral condition. **f** Population result of firing rate during the pre-choice period (all the correct trials in 1 step and 2 steps conditions were combined). Box-and-whisker plots indicate the minimum, 25th, 50th, 75th percentiles, and maximum excluding outliers (i.e., 1.5 times greater than the interquartile range). *$P = 4.8 \times 10^{-7}$; Mann–Whitney test (two-sided; $n = 594$ and $n = 306$ single-units for saline and CNO conditions, respectively. Shaded bands, 95% confidence intervals (**b**–**e**). Source data are provided as a Source Data file.

activity while the animals were performing the CAS task (CNO or saline control conditions) (Fig. 5a). We obtained 900 single-units in 43 sessions from five rats (594 and 306 units in saline and CNO conditions, respectively). Without CNO, some neurons showed activity that was selective to a specific rule (i.e., 1 step or 2 steps rule) even before the animal made 1st choice responses (Fig. 5b, c and Supplementary Fig. 12a, b. Also see Fig. 5d, e for an example single-unit activity with an IP injection of CNO). We calculated the mean firing rate of M2 single-units during a 1 s period immediately before animals made their 1st choice (i.e., pre-choice period) and found that chemogenetic suppression of ACC neural activity decreased firing rate both in ipsilateral and contralateral choice conditions (Fig. 5f).

We then quantified rule selectivity during the pre-choice period for each single-unit using receiver operating characteristic (ROC) analysis[29], which measures the degree of overlap between two response distributions[30,31]. For each M2 single-unit the preferred and non-preferred rule conditions were compared, given two distributions of neuronal activity (see Methods). An ROC curve was then generated by taking the observed firing rate of a neuron and then the area under the ROC curve was calculated. A value of 0.5 indicates that the two distributions were completely overlapped, and thus the neuron is not selective to the rules. A value of 1.0, on the other hand, indicates that the two distributions were completely separated and so the neuron is very selective. Time course of rule selectivity of the representative M2 single-unit showed an increase in rule selectivity in the pre-choice period in the ipsilateral choice condition but not in the contralateral condition (Fig. 6a, b; also see Supplementary Fig. 12c, d). We repeated the same analysis for all the M2 single-units that exhibited a mean firing

rate greater than 3 Hz during pre-choice period in either ipsilateral or contralateral conditions (437 and 195 single-units for saline and CNO conditions, respectively) (see Methods). Population-averaged time course of rule selectivity showed that chemogenetic silencing of ACC decreased rule selectivity of M2 neurons in ipsilateral choice trials and this tendency could be seen not only after animals made an ipsilateral choice but even before making the choice (Fig. 6c). Such an effect was not seen in contralateral conditions (Fig. 6d). We split the Rule Switch-blocks into three epochs and examined rule selectivity in each epoch for both CNO and saline conditions. The administration of CNO decreased rule selectivity in trials in which animals made a 1st response to the ipsilateral side followed by a 2nd response to the contralateral side (Fig. 6e, top left panel), while such a tendency was not observed in trials in which animals made their 1st response to the contralateral side followed by the 2nd choice to the ipsilateral side (Fig. 6e, bottom left and bottom right panels). The effect observed in the ipsilateral condition was greatest in the 1st epoch that immediately followed rule switches from the 1 step to 2 steps rules (Fig. 6e, top left panel). A similar result was obtained when we evaluated M2 neurons' rule selectivity using another ROC measure in which the distributions of firing rates of M2 neurons were matched between saline and CNO conditions (Supplementary Fig. 13; see Methods for details).

To test if a perturbation of ACC inputs to M2 neurons during pre-choice period could affect an animal's sequential choice performance in trials following rule switches, we optogenetically excited ACC terminals in M2 specifically during pre-choice period. Indeed, such perturbation affected 2nd choice performance (i.e., increased 2nd choice omission errors) (Supplementary Fig. 14). Strikingly, this effect

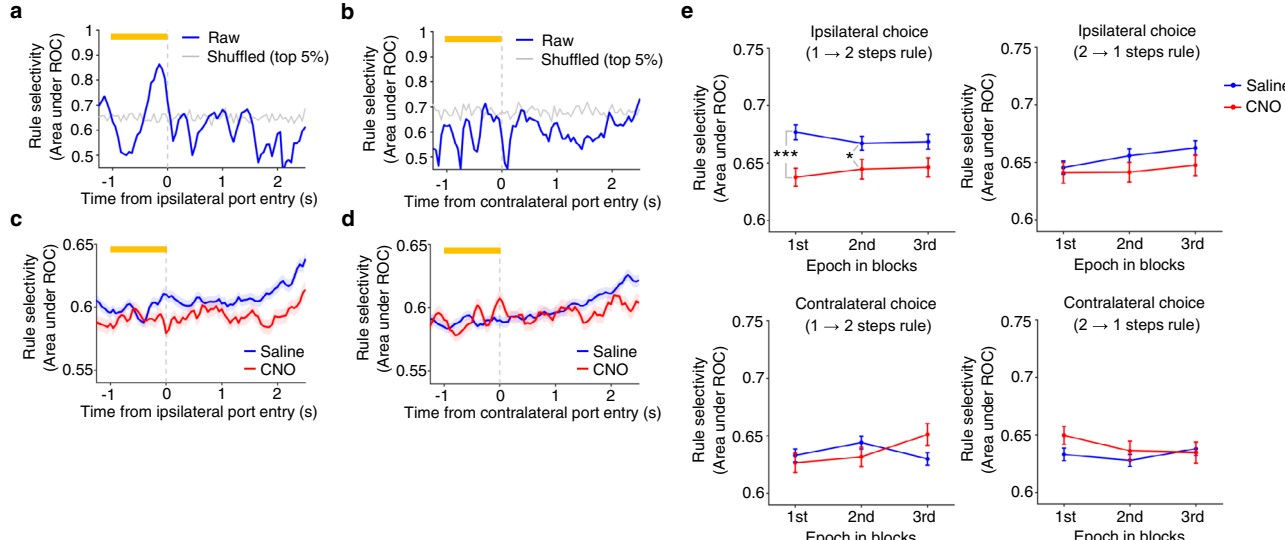

**Fig. 6 | Chemogenetic silencing of ACC decreased rule selectivity in M2 neurons. a** Time course of rule selectivity of a representative M2 single-unit (the same single-unit shown in Fig. 5b, c) for ipsilateral condition. Rule selectivity was quantified by applying ROC analysis to the distributions of mean firing rates during pre-choice period in trials of Rule Switch-blocks of 1 step and 2 steps conditions (see Methods for details). Gray line represents a 95% percentile level estimated by shuffled data in which we randomly shuffled rule labels for trials (1 step or 2 steps conditions) before calculating the area under ROC curve. Error bar, SEM. **b** Same as in **a**, but for contralateral condition. **c** Population-averaged time course of rule selectivity in ipsilateral condition. Blue, saline solution. Red, CNO solution. Shaded bands, SEM. **d** Same as in **c**, but for contralateral condition. Shaded bands, SEM. **e** Comparison of rule selectivity between CNO and saline conditions and across three epochs in Rule Switch-blocks (1st epoch, 1–18th trials; 2nd epoch, 17–36th trials; 3rd epoch, 37–55th trials). See Methods

for details of calculating rule selectivity in each epoch. A repeated measures two-way ANOVA (with epoch being a within-subject factor) was conducted for the ipsilateral condition with rule switches from 1→2 steps conditions. No interaction was found between CNO dose and epoch in the ipsilateral choice (1→2 steps rules) condition ($P > 0.4$). A significant main effect of CNO dose was detected ($F_{1,1833} = 12.7$, $P = 3.7 \times 10^{-4}$), but not for epoch ($P > 0.7$). Post hoc comparison using two independent samples $t$-test showed significant differences between saline and CNO solutions in 1st and 2nd epochs. No significant interaction or main effect was detected for other three conditions (i.e., ipsilateral condition with rule switches from 2→1 steps conditions, contralateral condition with rule switches from 1→2 steps conditions, and contralateral condition with rule switches from 2→1 steps conditions). ***$P = 4.4 \times 10^{-4}$. *$P = 0.036$. $n = 437$ and $n = 195$ single-units for saline and CNO conditions, respectively. Error bar, SEM. Source data are provided as a Source Data file.

was observed specifically in trials immediately following rule switches from 1 step to 2 steps rules, i.e., 1st epoch of 2 steps rule blocks (Supplementary Fig. 14b). Combined with the results from chemogenetic perturbations on M2 activity (Fig. 6), these results suggest that ACC modulates rule representations in M2 before an animal makes sequential choice responses in trials following rule switches from 1 step to 2 steps conditions.

**Chemogenetic silencing of ACC increased the firing rate of negative outcome-encoding M2 neurons upon rule switches**
In CAS task, no explicit cue signal for a change of rule was delivered before an animal completes the first trial following a change of rule and, instead, rule changes were delivered to an animal by outcome feedback signals (i.e., an omission of reward and a feedback buzzer sound). Therefore, animals were expected to adjust their 2nd choice responses based on the outcome feedback information. To address whether and how ACC circuits provide M2 neurons with outcome feedback information upon rule switches, we examined M2 activity during the outcome feedback period (594 and 306 single-units for saline and CNO conditions, respectively) (see Methods). We first tested if an M2 neuron showed a greater or smaller mean firing rate during the outcome feedback period than that during the baseline period and conducted several analyses using the database of such neurons. A chemogenetic silencing of ACC neurons decreased the proportions of neurons that were activated either during the positive outcome feedback period (i.e., positive outcome-activated neurons) or during the negative outcome feedback period (i.e., negative outcome-activated neurons) (Fig. 7a left and right panels, respectively). In contrast, a chemogenetic silencing of ACC neurons increased the proportions of neurons that were suppressed during the positive outcome feedback period (i.e., positive

outcome-suppressed neurons) and a similar tendency was observed for neurons that were suppressed during the negative outcome feedback period (i.e., negative outcome-suppressed neurons) although it did not reach a statistical significance (Fig. 7a left and right panels, respectively). These results showed that chemogenetic silencing of ACC neurons decreased the proportion of outcome-activated neurons and increased that of outcome-suppressed neurons irrespective of whether the outcome feedback was positive or negative.

We then plotted population-averaged PETHs of outcome-related M2 neurons for the 1st epoch (consisting of trials immediately following rule switches) and 2nd/3rd epochs (Fig. 7b). We found no qualitative difference between the 1st epoch and 2nd/3rd epochs in the time courses of positive outcome-activated neurons (Fig. 7b, top left), of positive outcome-suppressed neurons (Fig. 7b, top right), and of negative outcome-suppressed neurons (Fig. 7b, bottom right). On the other hand, negative outcome-activated neurons showed a qualitative difference in the time course of 1st epoch and 2nd/3rd epochs in CNO conditions (Fig. 7b, bottom left). To quantify this difference, we next compared the mean firing rates of negative outcome-activated neurons during a 3 s period following the delivery of error feedback tone between 1st epoch and 2nd/3rd epochs in CNO and saline conditions (Fig. 7c). These neurons showed a greater mean firing rate in 1st epoch than in 2nd/3rd epochs in CNO condition, while no such difference was found in saline condition. Such an effect was observed neither during the error feedback period following animals' incorrect 1st choices nor during the positive outcome period following animals' correct 2nd choices in Rule Switch-blocks (Supplementary Fig. 15a, d for incorrect 1st choices and Supplementary Fig. 15b, e for correct 2nd choices). Moreover, no such effect was observed during the error feedback period following animals' incorrect 2nd choices in 1st blocks

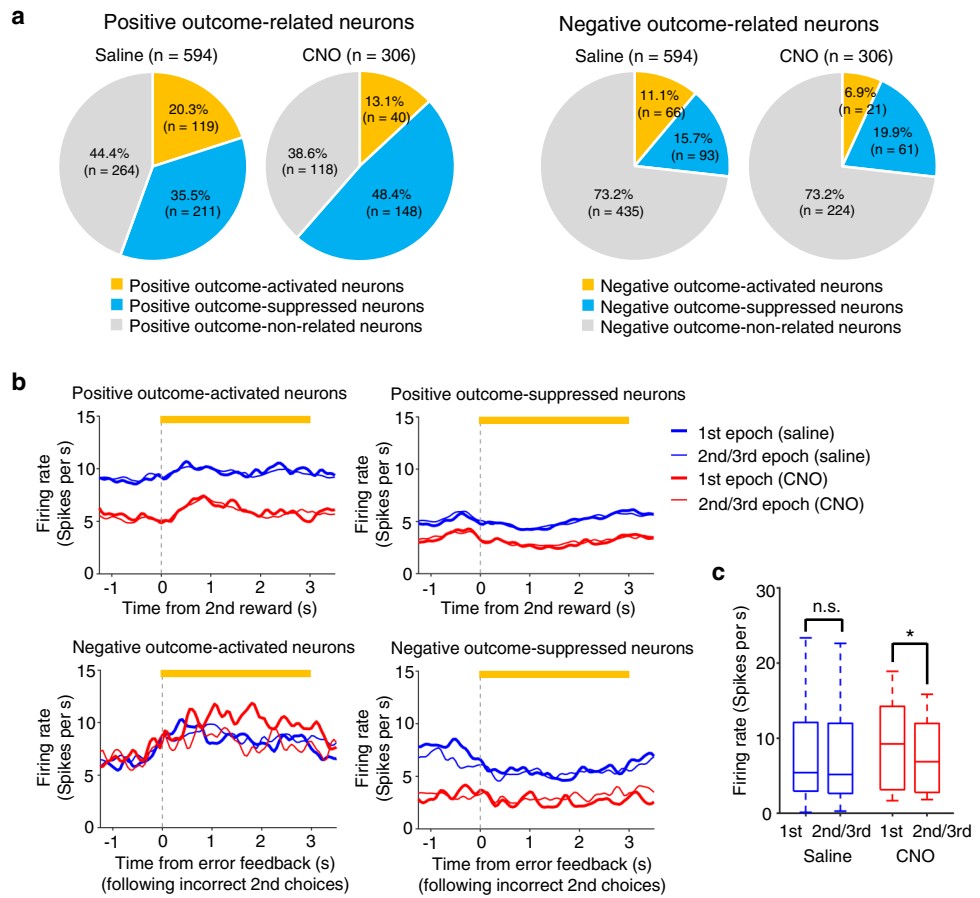

**Fig. 7 | Chemogenetic silencing of ACC increased firing rate of negative outcome-encoding M2 neurons upon rule switches. a** Each M2 neuron ($n = 594$ neurons and 306 neurons for saline and CNO conditions, respectively) was classified based on its activity during the outcome feedback period, i.e., positive outcome-activated neurons, positive outcome-suppressed neurons, negative outcome-activated neurons, and positive negative-suppressed neurons (see Methods for details). The proportion of positive outcome-activated neurons was smaller in CNO condition than in saline condition (13.1% vs 20.3%, $\chi^2 = 6.73$, $P = 0.009$) while that of positive outcome-suppressed neurons was greater in CNO condition than in saline condition (48.4% vs 35.5%, $\chi^2 = 13.9$, $P = 0.00019$). Similarly, the proportion of negative outcome-activated neurons was smaller in CNO condition than in saline condition (6.9% vs 11.1%, $\chi^2 = 4.17$, $P = 0.041$) while that of negative outcome-suppressed neurons was smaller in CNO condition than in saline condition but was not statistically significant (19.9% vs 15.7%, $\chi^2 = 2.61$, $P = 0.106$). **b** Population-

averaged PETHs of positive outcome-activated neurons (top left, $n = 66$), positive outcome-suppressed neurons (top right, $n = 66$), negative outcome-activated neurons (bottom left, $n = 19$), negative outcome-suppressed neurons (bottom right, $n = 21$). Red, CNO (IP, 20 mg/kg). Blue, saline. Thick and thin lines represent 1st and 2nd/3rd epochs in 2 steps condition, respectively. **c** Mean firing rate of each negative outcome-activated neuron was calculated for a 3 s period starting from the onset of error feedback tone presentation in incorrect 2nd choice trials, and was compared between 1st and 2nd/3rd epochs in Rule Switch-blocks of 2 steps condition. A significant difference was found between 1st and 2nd/3rd epochs in CNO condition ($n = 19$ neurons; median, 9.23 spikes s$^{-1}$ vs 6.86 spikes s$^{-1}$; *$P = 0.018$, two-sided signed rank test) while no difference was found in saline condition ($n = 66$ neurons; median, 5.41 spikes s$^{-1}$ vs 5.17 spikes s$^{-1}$; n.s., $P = 0.53$, two-sided signed rank test). Box-and-whisker plots indicate the minimum, 25th, 50th, 75th percentiles, and maximum. Source data are provided as a Source Data file.

(Supplementary Fig. 15c, f). Taken together, these results showed that a chemogenetic silencing of ACC excessively increased the activity of negative outcome-activated M2 neurons specifically during the negative outcome feedback period following animals' incorrect 2nd choices upon rule switches from 1 step to 2 steps conditions. The enhancement of the activity of negative outcome-activated neurons was specifically observed in the 1st epoch of Rule Switch-blocks and is consistent with the behavioral data in this respect while other outcome-encoding neurons did not show epoch-specific effect of ACC silencing. Therefore, it seems likely that the epoch-specific disruption of animals' 2nd choice performance is associated with the excessive enhancement of the activity of negative outcome-activated neurons caused by the ACC silencing (also see Discussion section).

**Optogenetic silencing of ACC circuits induced errors in trials immediately following 1 step to 2 steps rule switches**
To test if suppression of ACC circuits during the outcome period could affect animals' sequential choice performance as was suggested by the

chemogenetic modulation of activities of outcome-related neurons in M2 (Fig. 7), we optogenetically suppressed ACC excitatory neurons using halorhodopsin (eNpHR3.0) (Fig. 8a). A 561-nm laser light delivery decreased spiking activity of ACC neurons (Fig. 8b), which validated in vivo neuronal inhibition.

We delivered the light for 4 s immediately following the animals' incorrect 2nd choices (Fig. 8c) or the animals' correct 2nd choices (Fig. 8d). In a representative session, the light was delivered upon animals making incorrect 2nd choices. The animal showed increased error in 2nd choices in trials following light delivery (Fig. 9a). Group results demonstrated that optogenetic suppression after the animal's erroneous 2nd choices induced such errors in 2nd choices in the trials that followed initial errors (Fig. 9b). This effect was specifically observed in the 1st epoch of Rule Switch-blocks but not in other epochs in these blocks or any epoch in the 1st block. Also, the error rate of 1st responses in these animals was unaffected, suggesting that the animals failed to update their sequential choice responses due to ACC inhibition (Fig. 9d).

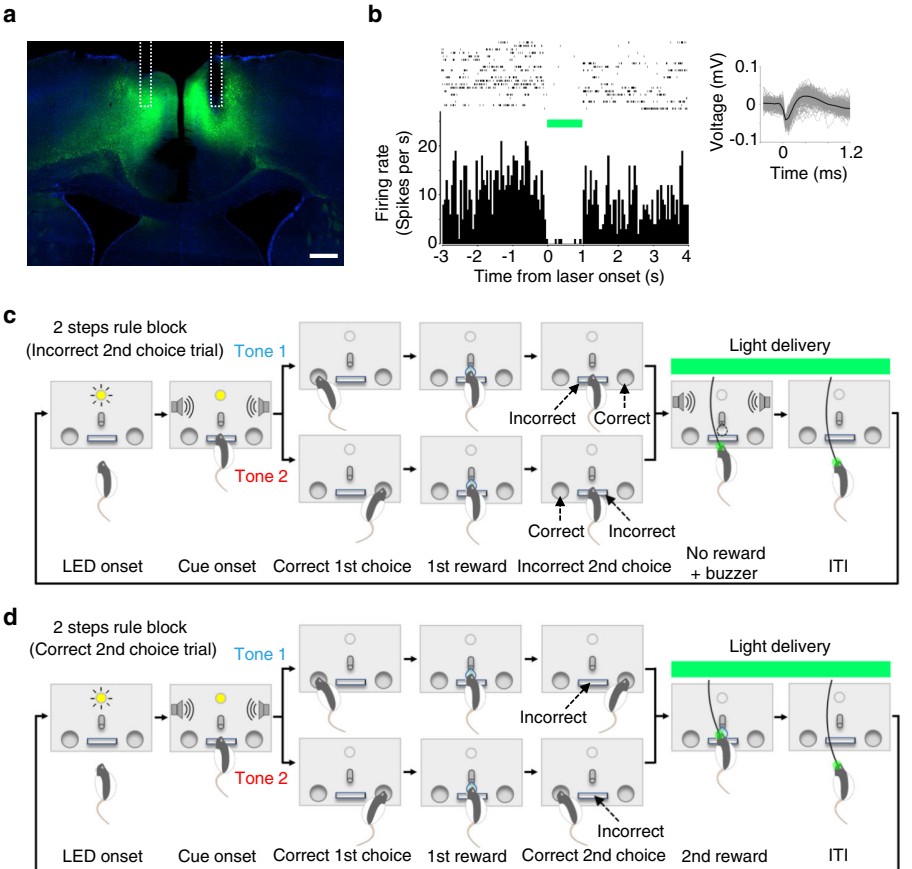

**Fig. 8 | Optogenetic silencing of ACC neurons during outcome feedback period following animals' 2nd choices. a** Histological section for halorhodopsin (eNpHR3.0) expression in ACC. Green, eNpHR3.0-eYFP expression. Blue, DAPI. White dotted line shows reconstructed positions of fiberoptic implants. Scale bar, 500 μm. **b** Suppression of spiking activity by 561 nm light delivery. Top left, raster plot of a representative single-unit measured in ACC showing spiking activities before, during, and after laser light delivery. Top right, example waveforms of the representative single-unit. Bottom left, peri-event time histogram sorted by the timing of light onset. Bin width, 50 ms. **c** Optogenetic silencing of ACC after animal's incorrect 2nd choices (i.e., 2nd choice omission errors). Light was delivered for 4 s after animals pushing the center lever instead of correctly poking the side port opposite to the 1st choice. **d** Optogenetic silencing of ACC after animal's correct 2nd choices. Light was delivered for 4 s after a correct 2nd choice (i.e., animals poking the side port opposite to the 1st choice). Source data are provided as a Source Data file.

We repeated similar experiments with 4 s light applied immediately after the animals' correct 2nd choices (Fig. 8d). Light delivery did not affect 2nd choice performance in any epoch of the 1st or Rule Switch-blocks (Fig. 9c). Also, the error rate of 1st responses was unaffected (Fig. 9e). The observed effect of optogenetic silencing after the animals' incorrect 2nd choice on their task performance (Fig. 9b) could not be explained by the heat of light because the same duration of light delivery did not affect the task performance when the light was delivered after the animals' correct 2nd choices (Fig. 9c). These results indicate that ACC neurons process error feedback information following an erroneous 2nd response and use this information to adjust the animal's sequential choice responses in subsequent trials.

Finally, we tested if an optogenetic excitation of ACC inputs to M2 neurons during the outcome period could affect an animal's sequential choice performance (Supplementary Fig. 16a, b). Contrary to the results from optogenetic silencing of ACC neurons (Fig. 9), optogenetic excitations of ACC-M2 projections using ChR2 during the outcome feedback period did not affect the 2nd choice performance in the immediately following trials (Supplementary Fig. 16c, d). Similarly, such perturbation did not affect the 1st choice performance in the immediately following trials (Supplementary Fig. 16e, f).

## Discussion
This study found that chemogenetic silencing of ACC neurons in behaving rats increased errors in sequential choices, decreased rule

selectivity of M2 neurons' activity during the pre-choice period, and enhanced the activity of negative outcome-activated M2 neurons during the 2nd choice outcome feedback period. These effects were observed specifically in trials immediately following rule switches, suggesting a critical role of ACC-M2 circuits in flexibly updating neural representations of task rules. Previous studies have suggested that anterior cingulate circuits are recruited when a greater cognitive control is needed for an animal to resolve uncertain situations in which it experiences unexpected errors or conflicts among multiple choice options, needs to use error feedback for future decisions, or obtain new information for updating error likelihood[15–18,20,32–34]. While this study generally comports with these studies regarding the role of ACC circuits in performance monitoring, this study examined neural activity in the downstream M2 cortex and could reveal the roles of ACC-M2 pathway in flexible updating of rule representations upon rule switches.

What is the functional role of ACC-M2 pathway in an animal's sequential decision making and rule switching? Do ACC inputs just increase the excitability/activity of M2 neurons in a way that enables them to function normally or do they actually convey task-related information? If the latter is the case, what information do ACC-M2 projections convey? A previous rat study showed that inhibiting M2 neurons caused decrease of left/right binary and single-step choice performance, suggesting that modulating M2 activity per se affects single-step choice performance[8]. In contrast, we showed that

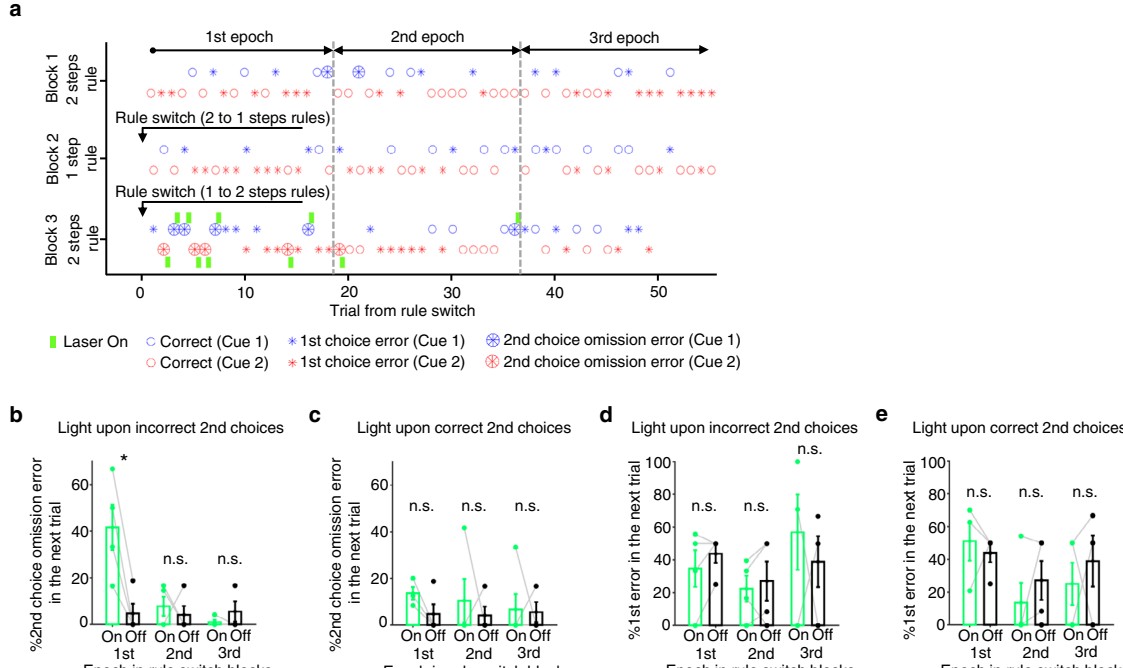

**Fig. 9 | Optogenetic silencing of ACC neurons upon incorrect 2nd choices induced sequential choice errors in the immediately subsequent trials that followed rule switches. a** Task performance chart of a representative session in which light was delivered upon incorrect 2nd choices in Rule Switch-block. Green bar, light delivery. Small circle, correct trial. Asterisk, 1st choice error. Large circle filled with an asterisk, 2nd choice omission error. Blue and red represent two distinct tone cues. Gray dotted lines show borders separating three epochs in each block (1–18th, 19–36th, and 37–55 trials for 1st, 2nd and 3rd epochs, respectively). **b** 2nd choice performance was plotted for trials that immediately followed a 2nd choice omission error trial. On and Off represent trials in which light was delivered and not delivered, respectively. The performance was plotted for each of the three epochs separately. *$P = 0.0201$ for 1st epoch, $P = 0.575$ and $0.391$ for 2nd and 3rd epochs, respectively (two-sided two samples $t$-test, $n = 4$ rats). Error bar, SEM.

**c** Light was delivered for 4 s after correct 2nd choices instead of incorrect 2nd choices (see Fig. 8d). 2nd choice performance was plotted for trials that immediately followed a 2nd choice omission error trial. $P = 0.544$, $0.780$ and $0.641$ for each epoch, respectively. Two samples $t$-test (two-sided), $n = 4$ rats. Error bar, SEM. **d** Same as in **b**, but 1st choice performance was plotted for trials that immediately followed a 2nd choice omission error trial in which a light was delivered upon incorrect 2nd choices (On condition) or not delivered (Off condition). $P = 0.212$, $0.598$ and $0.913$ for each epoch, respectively. Two samples $t$-test (two-sided), $n = 5$ rats. Error bar, SEM. **e** Same as in **c**, but 1st choice performance was plotted for trials that immediately followed a 2nd choice omission error trial in which light was delivered upon correct 2nd choices (On condition) or not delivered (Off condition). $P = 0.640$, $0.503$ and $0.587$ for each epoch, respectively. Two samples $t$-test (two-sided), $n = 5$ rats. Error bar, SEM. Source data are provided as a Source Data file.

suppressing ACC-M2 projection affected 2nd choice performance without affecting the 1st choice performance. This distinction between our study and previous study suggests that ACC-M2 projections do not just increase the activity level of M2 neurons but specifically promote the performance of a forthcoming sequential choice response. Furthermore, the effect of suppressing ACC-M2 projection on an animal's 2nd choice performance was specifically seen in trials immediately following rule switches (i.e., 1st epoch in 2 steps rule blocks), suggesting that the role of ACC-M2 projection is not limited to performing an extra step but rather is linked to flexibly adjusting the performance of an additional step in an action sequence upon rule switches.

How does the rule selective activity of M2 neurons during the pre-choice period relate to animals' behavior? Previous study has shown that neurons in premotor and supplemental motor areas can encode planning for several movements ahead[5]. Given that a chemogenetic silencing of ACC-affected rule selectivity of M2 neurons during pre-choice period in trials that immediately followed rule switches (Fig. 6), it is likely that ACC generates a signal required for reorganizing plans of sequential choice actions that are represented in neurons in downstream motor areas. Indeed, an optogenetic excitation of ACC terminals in M2 during pre-choice period (Supplementary Fig. 14) increased 2nd choice omission errors in the next trials and this effect was specifically observed in trials immediately following rule switches from 1 step to 2 steps conditions, supporting the idea that ACC provides M2 neurons with information that is useful for making forthcoming sequential choices in the new rule condition.

How does the outcome-encoding activity of M2 neurons relate to animals' behavior? We found distinct effects of ACC silencing on outcome-encoding activity in M2 neurons (Fig. 7). When ACC neurons were chemogenetically silenced, the proportion of outcome-activated neurons decreased while that of outcome-suppressed neurons increased irrespective of whether the outcome feedback was positive or negative. The reduction of proportions of outcome-activated M2 neurons may affect an animal's capacity to use outcome information (either positive or negative) for rule updating in some ways and this could affect animals' 2nd choice performance. Importantly, the epoch-specific effect of ACC silencing was observed only in negative outcome-activated neurons and so other outcome-encoding neurons cannot explain the epoch-specific disruption of animals' 2nd choice performance in a simple manner. It might be that a decrease in the proportion of positive outcome-activated M2 neurons may attenuate an animal's capacity to use positive outcome information for maintaining the representation of the currently working rule throughout the block. In contrast, a decrease in the proportion of negative outcome-activated M2 neurons may attenuate an animal's capacity to use negative outcome information for updating rule representation upon rule switches. Another consideration besides the proportion statistics is the dynamics of outcome coding activity in M2 neurons (Fig. 7b). When ACC was chemogenetically silenced, the activity of negative outcome-activated M2 neurons increased specifically in trials immediately following rule switches (i.e., 1st epoch) while that of other types of outcome-encoding neurons decreased irrespective of the

epochs (Fig. 7b, c), suggesting distinct circuit mechanisms with which ACC could affect the activity of M2 neurons (Fig. 10a, b). Then, how can such distinct effects of ACC silencing on outcome-encoding M2 neurons relate to animals' rule switching behavior? Evidence accumulation models may provide some clues to address this question (Fig. 10c, d). In one model, ACC silencing decreases the step size of evidence accumulation for rule updating, thus requiring more evidence accumulation (i.e., outcome feedback) before the evidence crosses the threshold (Fig. 10c). In another model, ACC silencing pushes up the threshold for rule updating, requiring more evidence accumulation for threshold crossing (Fig. 10d). Which of these models is more likely considering our neural activity data? The observation that silencing of ACC neurons increased the activity of negative outcome-activated M2 neurons (Fig. 7b, c) suggests that this increased activity may be associated with pushing up the evidence threshold. On the other hand, the observation that silencing of ACC neurons decreased the activity of other outcome-encoding M2 neurons suggests that this decreased activity may be associated with a decrease in the step size. Given that the enhancement of the activity of negative outcome-activated neurons was specifically observed in the 1st epoch of Rule Switch-blocks and is consistent with the behavioral data in this respect, it seems more likely that the epoch-specific disruption of animals' 2nd choice performance caused by ACC silencing (Figs. 2, 4, and 9) reflected the excessive increase of activity of negative outcome-activated neurons (Fig. 7b, c) and its pushing up the evidence threshold for rule updating (Fig. 10d). Note that the epoch-specific effect of ACC silencing was

observed only in negative outcome-activated neurons and so other outcome-encoding neurons cannot explain the epoch-specific disruption of animals' 2nd choice performance in a simple manner. Taken together, based on the evidence accumulation models of rule updating, the neural activity data suggests a possibility that ACC silencing pushed up the threshold for rule updating by enhancing the activity of negative outcome-activated M2 neurons upon rule switches, and this excessive activity led to the epoch-specific disruption of animals' 2nd choice performance (Fig. 10d).

We observed an effect of chemogenetic silencing of ACC circuits in updating rules from 1 step to 2 steps but not in the opposite direction (i.e., rule switches from 2 steps to 1 step) as measured by non-rewarded second actions. This asymmetric effect of chemogenetic silencing may indicate that ACC circuits are recruited in rule switches that demands an increment but not a decrement of response actions. Given that making a 2 steps choice would demand a greater cost on the part of the animals compared to making a single-step response, the asymmetric effect (an increase of 2nd choice omission errors in animals with chemogenetic silencing of ACC circuits) raises a possibility that ACC circuits are recruited specifically in task settings that demand an increased effort/cost for making choice responses[35]. Alternatively, this asymmetric effect may reflect some inherent network characteristics in ACC→M2 circuits such as distinct temporal scales of transitions of population neural activity in M2[36]. Previous studies reported that some subpopulations of dorsal striatal neurons and their downstream circuits are recruited for completing an action

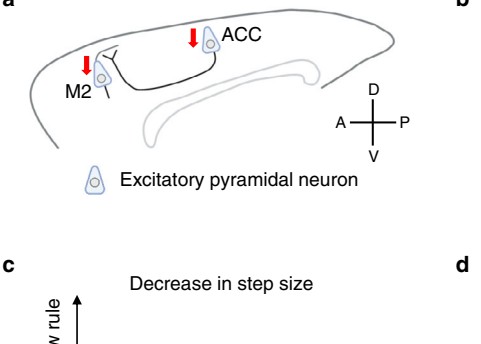

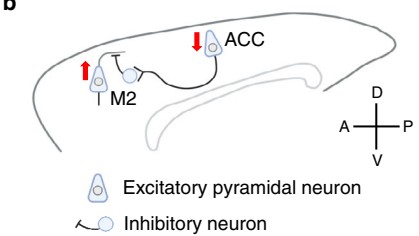

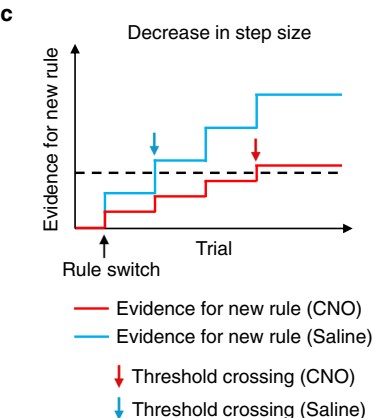

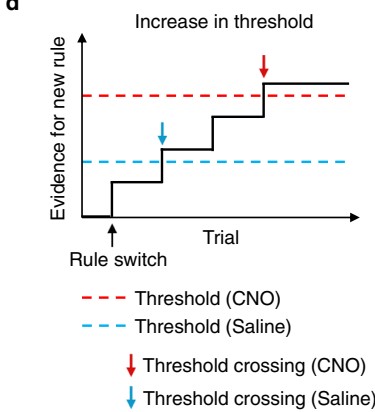

**Fig. 10 | Possible mechanisms of how silencing of ACC neurons affects animals' task-switching performance by modulating rule representations in M2 circuit.** **a** A possible ACC-M2 circuit in which silencing of ACC neurons decreases the activity of outcome-encoding M2 neurons (positive outcome-activated neurons, positive outcome-suppressed neurons, and negative outcome-suppressed neurons; see Fig. 7b for their activity profiles). Red down arrow indicates a decrease in activity. **b** A possible ACC-M2 circuit in which silencing of ACC neurons enhances the activity of negative outcome-activated M2 neurons (see Fig. 7b, c for its activity profile). In this circuit model, when ACC neurons are silenced, a disinhibition mechanism in M2 increases the activity of negative outcome-activated M2. Red upward or downward arrow indicates an increase or a decrease in activity, respectively. **c** An evidence accumulation model describing how silencing of ACC neurons can affect an animal's rule switch performance. In this model, an animal updates the choice rule (i.e., 1 step or 2 steps rule) when the evidence for the new

rule crosses a certain threshold level. The evidence for the new rule can increase stepwise every time the animal receives an outcome feedback after the animal makes 2nd choice. When ACC neurons are not silenced, after the animal experiences several error trials upon rule switches, the evidence for the new rule can cross the threshold, enabling the animal to make a rule switch (blue). In contrast, when ACC neurons are silenced, the step size of evidence accumulation for the new rule decreases, thus requiring more evidence accumulation (i.e., outcome feedback) before it crosses the threshold (red). Down arrows represent trial positions at which the evidence for the new rule crosses the threshold (blue, saline; red, CNO). **d** Format is the same as in **c**, but, in this model, instead of decreasing the step size of evidence accumulation, a silencing of ACC neurons pushes up the threshold for rule updating, thus requiring more evidence accumulation for threshold crossing. Blue and red dotted lines, threshold for rule updating in saline and CNO conditions, respectively.

sequence or extinguishing an action[13,37]. By analogy, updating rules from 2 steps to 1 step which requires deletion of the second step response might depend on striatal or its downstream circuits rather than ACC→M2 circuits. Given that more global brain networks are recruited when an animal makes movements that are not instructed by the behavioral task and are not rewarded[38], brain networks beyond ACC→M2 circuits (cortical or subcortical) might be necessary to extinguish such non-rewarded movements upon rule switches.

Chemogenetic silencing of ACC neurons increased 2nd choice omission errors in the first epoch of 2 steps condition and, on average, this effect emerged toward the end of the first epoch (Figs. 2i and 4f). Ideally, animals should be able to make rule switching in the next trial after the animal makes an inevitable error due to the rule switch. However, in reality, animals seem to need some trials to abolish the old rule and to switch to the new one even in normal condition (i.e., no silencing of ACC) and this seems to be the reason why we did not see a significant difference between saline and CNO conditions in animals' 2nd choice performance in the earlier part of the first epoch. But, first of all, why do animals need more than one error trial to make rule switches? In evidence accumulation models of rule switching (Fig. 10c, d), if the single step size of evidence accumulation is smaller than the threshold for rule updating, animals are expected to require multiple trials to accumulate the evidence for the new rule before the evidence crosses the threshold, irrespective of whether ACC neurons are silenced or not. In such condition, even if ACC-M2 circuits can maintain and update knowledge regarding what actions would maximize the rewards and thus can be a neural resource for model-based choice decisions, animals may still need several trials before they can fully utilize such knowledge in the face of rule switches. It remains unexplored whether a longer task training or any other task arrangement could make the step size for evidence accumulation greater and enable an animal to make a one-shot rule updating.

A chemogenetic silencing of ACC neurons or their terminals in M2 decreased 2nd choice performance by around 20% on average but did not completely abolish animals' ability to update their choice responses upon rule switches (Figs. 2 and 4). It might be because a chemogenetic silencing could suppress not a whole but only a partial proportion of ACC neuronal activities (Supplementary Fig. 2). With rich recurrent connections, the ACC circuit or its downstream circuits in M2 may compensate for the reduced neuronal activities in ACC and reproduce a robust neural computation underlying an animal's sequential choice decisions. Alternatively, brain mechanisms other than ACC-M2 circuits may also take part in the flexible updating of sequential choice responses upon rule switches. Such mechanisms might coexist and work in parallel with ACC circuits in outcome monitoring, and/or updating and preparing for upcoming sequential choice responses. It might be that ACC-M2 circuits maintain, update and utilize knowledge regarding what actions would maximize the rewards in the face of rule switches of the task (e.g., model-based choice decisions) while other brain mechanisms may complement ACC-M2 circuits by promoting exploration strategies with which animals may find more rewarding choices without requiring any knowledge regarding what choice responses maximize rewards upon rule switches (e.g., model-free choice decisions)[39]. It is an open question whether and how such processes could be recruited in distinct ways for flexibly updating sequential choice responses upon changes of rules in the environment.

Interestingly, we found no effect of chemogenetic silencing of prelimbic/infralimbic cortex (presumably phylogenetic homolog of dorsolateral prefrontal cortex in primates) while previous studies have shown that DLPFC also plays a critical role in updating rule representations in the brain[19,30,40]. In this study, we minimized the within-trial working memory load while requiring animals to hold rule representations (e.g., rule memory or task set) across trials in specific rule blocks. The cognitive loads of different time scales (i.e., memory spanning a single trial period is several seconds vs. across rule blocks is ~10 min) might affect how an update of behavioral response would rely on DLPFC vs. ACC circuits[32]. Alternatively, the reason might be due to the difference in the characters of the task rule that we focused on in this study: in previous rodent studies that showed the involvement of prelimbic/infralimbic cortices in rule switching, these were related to choices in space (e.g., turning left or right in a maze) while, in this study, we tested animals' ability of updating internal representations of rules related to response sequences (i.e., switching between 1 step response and 2 steps response)[41,42]. It is an open question what distinct roles ACC and prelimbic/infralimbic circuits play in flexibly adjusting responses upon rule switches.

An optogenetic excitation of ACC-M2 projections during the outcome period did not affect animals' choice performance while an optogenetic suppression of ACC neurons during the same task period could affect it (Fig. 9 and Supplementary Fig. 16). This discrepancy raises several possibilities. First, the expression of ChR2 in ACC neuronal axons terminating in M2 may not have been strong enough to elicit a behavioral effect. Second, the spread of excitation light at M2 may not have been broad enough and thus could not excite large portions of ChR2-expressing ACC axon terminals. However, considering that the same stimulation protocol could affect the task performance when the light was delivered during the pre-choice instead of the outcome feedback period (Supplementary Fig. 14), these two possibilities are not very likely. Alternatively, it is possible that the ChR2 stimulation protocol used in this study was not optimal. The observation that a chemogenetic silencing of ACC neurons did not decrease but increased the activity of negative outcome-activated M2 neurons (Fig. 7b, c) suggests an involvement of a disinhibition mechanism (Fig. 10b). With such a mechanism, a ChR2 stimulation at ACC terminals in M2 may affect animals' performance in an epoch-dependent manner if it drives sufficiently large portions of inhibitory M2 neurons that receive ACC afferents (note that, among the four types of outcome-encoding M2 neurons, only negative outcome-activated neurons showed an epoch-dependent effect of ACC silencing; see Fig. 7b, c). While we used a relatively short pulse duration in ChR2 excitation (i.e., 5 ms) which was previously shown to be effective for exciting fast-spiking interneurons, the pulse frequency (10 or 20 Hz) used in this study is likely to be suboptimal for driving these neurons[43]. Given these considerations, while we found no behavioral effect in ChR2 excitation experiments during the outcome period, a ChR2 excitation with a higher light pulse frequency can possibly affect animals' task performance. Including this issue, precise circuit mechanisms regarding how ACC-M2 circuits work in rule switching remain unexplored and it is an open question whether and how ACC neurons could modulate downstream M2 neurons possibly in cooperation with local inhibitory neuronal networks.

## Methods

We confirm that our research complies with U.S. National Institutes of Health guidelines and the Massachusetts Institute of Technology Department of Comparative Medicine and Committee of Animal Care (the approved protocol no., Tonegawa 0121-006-24).

### Subjects

Wild-type male Long-Evans rats (>300 g) were used (Charles River). Rats were pair-housed during initial behavioral training and then single-housed after being injected with viruses or being implanted with electrodes, fiberoptic implants, or cannulae. Rats were kept on a reverse 12 h light/dark cycle, and trained and tested in their dark cycle. Food was available ad libitum, and rats had scheduled access to water for motivating them to work for water reward while monitoring their body weight to ensure they were over 85% of their initial weight.

## Behavioral apparatus

Behavior took place in a custom-made chamber (415 mm length, 300 mm width, 500 mm height) inside a sound-attenuating cubicle (MED Associates). The cubicle was electromagnetically shielded by a copper mesh sheet or nickel/silver fabric in electrophysiology recording experiments. The behavioral setup consisted of a stainless-steel lever at the center (MED Associates) and two ports equipped with infrared photodiodes on the left and right sides of the lever, arranged side-by-side with a center-to-center distance of 65 mm on a stainless-steel wall. An interruption of the infrared beam signaled port entry. A sipper tube was installed on the front wall 25 mm above the center lever and was connected to a water supply that was controlled by a computer-controlled solenoid. In addition, there were two speakers mounted on the side walls (about 150 mm away from the center lever). Timing of presentations of sounds from the speakers (i.e., cue stimulus tones or feedback buzzer sound) and delivery of water rewards were controlled using a multifunction digital input/output board (NI USB-6343, National Instruments) with custom programs written in C++ (Visual Studio 2013, Microsoft) and LabVIEW (LabVIEW 2015 and Lab-VIEW FPGA Module 2016, National Instruments) on a computer running a Windows 10 operating system. Behavioral events were timestamped with a precision of <1 ms.

## Conditional action sequencing (CAS) task

**(1) 1 step condition in the CAS task**. Rats self-initiated each trial with a push on the center lever to receive the tone cue stimulus. After a delay of 10 ms, a tone of either 8 or 12 kHz with a sound pressure level of 75 dB was presented in pseudorandom order in each session. The tone was kept on until an animal entered one of the side ports (left or right) as the 1st choice. Choices were rewarded with ~25 μl water if they poked the correct side port. An error feedback buzzer sound was delivered as a penalty if they poked the incorrect side port followed by an elongated period of ITI. A feedback buzzer sound accompanied by an elongated ITI was also delivered if animals made a 2nd choice commission error, that is, entered the opposite side port after making a correct 1st choice and before initiating the next trial (pushing the center lever after an ITI). This completed a trial and, after an ITI period (3–4 s following a correct choice trial and 5–6 s following an incorrect choice trial), an LED turned on, signaling the animals to initiate the next trial by pushing the center lever.

**(2) 2 steps condition in the CAS task**. In the 2 steps condition, in addition to making the first-choice response to one of the side ports (depending on the tone cue), animals were required to make a 2nd response by entering the side port opposite the one that animals chose as the 1st response, instead of pushing the center lever. A correct entry to the side port as a 2nd response was rewarded with ~25 μl water. If animals pushed the center lever before making a correct 2nd response (i.e., an entry into the side port opposite to the 1st response), an error feedback buzzer sound was delivered. In this way, switches from 1 step to 2 steps rules were directly signaled to the rats: when the rule was switched from 1 step to 2 steps conditions and if the rats completes the 1st response (following the rule in the previous block, i.e., 1 step rule block), rats would push the center lever to initiate the next trial, causing an error buzzer feedback as an explicit feedback signal.

1 step and 2 steps conditions were switched in every 55 trials (with exceptions of 2 sessions in which rules were switched in every 40 trials) in a block-wise manner. In some sessions, behavior experiments started from the 2 steps condition (1st block) and then switched to the 1 step condition (2nd block) and so forth (see Supplementary Fig. 1a). In other sessions, experiments started from the 1 step condition in the 1st block so that animals' task performance in 1st block and Rule Switch-blocks in 1 step condition could be compared. No explicit cue signal for a change of rule was delivered before an animal completes the first trial following a change of rule and so there is no difference in the attentional loads to sensory inputs between 1 step and 2 steps conditions (note that an error would be inevitable in the first trials following rule changes because no explicit cue signal was delivered for a change of rule). Instead, rule changes were delivered to an animal by outcome feedback signals (i.e., an omission of reward and a feedback buzzer sound) in both task conditions (and so there is no difference in the attentional loads to sensory inputs from rewards and feedback buzzer tones between the two task conditions). In this task design, animals were expected to adjust their 2nd choice responses (poking the opposite side port or pushing the center lever to initiate the next trial without poking the side port) based on the outcome feedback information. In both directions of rule switches (i.e., from 2 to 1 step rule switch and the opposite direction), an exploration or any active perceptual processes other than perceptions of an omission of reward or feedback tones were not required on the part of the rats because these rule switches were directly signaled to the animals.

## Training

Rats were trained over the course of 8–12 weeks, with a progressive introduction of each aspect of the task as follows. After handling and habituation sessions in the behavior box (2–3 days), rats were first trained to either push the lever or enter a side port (left or right) to receive a water reward. Next, rats were trained to perform a contingency task in which a tone sound (4 kHz) was delivered when the animal pushed the center lever, and then the animal was required to poke either the left- or right-side port to receive a reward (typically required several sessions). Then rats were trained on auditory discrimination in which they were required to choose left- or right-side ports depending on the tone cue stimulus (either 8 or 12 kHz). Once they reached over 70% correct performance for two consecutive days, they were trained with the CAS task without a rule change between the 1 step and 2 steps conditions, requiring animals to make 2 step responses throughout the session. Once they reached over 70% correct performance for 2 consecutive days, they were trained on the CAS task with rule changes between the 1 step and 2 steps conditions once every 70–100 trials per block. The length of a block was gradually shortened to 55 trials to complete the task training phase. In many training sessions and in some testing sessions, we adjusted the proportion of two trial types (i.e., tone cues) to abate animals' choice bias to specific side ports. Such adjustments were typically limited to a ratio of less than 2.

## Surgery

All surgeries were performed under isoflurane anesthesia (1.0–2.0%) using the standard stereotactic technique. Following an IP administration of a cocktail solution of ketamine (80 mg kg$^{-1}$) and xylazine (8 mg kg$^{-1}$), rats were placed in an isoflurane induction chamber for 5–10 min. Then rats were moved to a stereotactic frame and their nose was placed in a cone, which provided 1.5–2% continuous isoflurane flow. After verifying surgical levels of anesthesia with pinch tests and eye blink tests, rats were secured in non-rupture ear bars (Kopf Instruments). The concentration of isoflurane was maintained at 0.75–1.5% throughout the surgery. Slow release buprenorphine (1 mg kg$^{-1}$) and Ringer's solution were administered in the middle and at the end of the surgery, respectively. Rats remained on a heating pad until they made a full recovery from anesthesia after surgery. Supplemental nutrient gels were supplied to rats after they were returned to their home cages. Rats were monitored during their recovery from surgery for at least 4 days before restarting experiments.

## Viral injections

For chemogenetics experiments (Figs. 2 and 4–7 and Supplementary Figs. 2–7, 9–13, and 15), we used AAV5-CaMKIIa-hM4Di-mCherry and AAV$_5$-CamKIIa-mCherry viruses (Addgene, Plasmid#50477 and Plasmid#114469). For optogenetic silencing of ACC neurons (Figs. 8 and 9),

we used AAV9-CamKIIa-eNpHR3.0-eYFP virus (Addgene, Plasmid#26971). For optogenetic excitation of ACC terminals in M2 cortex (Supplementary Figs. 14 and 16), we used AAV9-hSyn-hChR2(H134R)-eYFP virus (Addgene, Plasmid#26973). All the plasmids used in chemogenetics and optogenetics experiments were obtained from Addgene and were packaged by Vigene after in-house plasmid preparation. The viral titers were $3.3 \times 10^{14}$ genomic copy (GC) ml$^{-1}$ for AAV5-CamKIIa-hM4Di-mCherry, $1.9 \times 10^{13}$ GC ml$^{-1}$ for AAV5-CamKIIa-mCherry, $1.3 \times 10^{13}$ GC ml$^{-1}$ for AAV9-CamKIIa-eNpHR3.0-eYFP, and $1.1 \times 10^{13}$ GC ml$^{-1}$ for AAV9-hSyn-hChR2(H134R)-eYFP viruses. For viruses used in viral tracing experiments (Fig. 3 and Supplementary Fig. 8), see the following section for viral tracing experiments.

Each animal underwent bilateral craniotomies using a 1/4 size drill bit. The virus solutions with a volume of 1 µl were injected using a mineral oil-filled glass micropipette joined by a microelectrode holder to a 10 µl Hamilton micro syringe. A micro syringe pump was used to control the speed of virus injections (2 nl min$^{-1}$). The micropipette was slowly lowered to the target site and remained for 5 min before starting injections (ML ± 1.3 mm, AP + 2.0 mm, DV −0.9 mm for M2; ML ± 0.5 mm, AP −1.0 mm, DV −1.4 mm for ACC; ML ± 0.6 mm, AP + 3.0 mm, DV −3.0 mm for prelimbic/infralimbic cortex; ML ± 2.0 mm, AP −2.2 mm, DV −6.0 mm for ventral thalamic nuclei). After injections, the micropipette stayed for 10 min before it was withdrawn.

## Viral tracing experiments

For exploring brain regions projecting to M2 in the rat, we used a genetically engineered rabies virus[25]. We first unilaterally injected a cocktail solution of pENN.AAV.CaMKII.0.4.Cre.SV40 (Addgene#105558-AAV9) and AAVrh8-synP-DIO-sTpEpB-WPRE-bGH with a mixed ration of 1:1 at M2 (1 µl, ML ± 1.3 mm, AP + 2.0 mm, DV −0.9 mm)[44]. One to two weeks later, RVΔG-4mCherry (EnvA) (1 µl) was injected at the same coordinates[45]. Viral titers for the injected solution were $1.05 \times 10^{13}$ GC ml$^{-1}$ and $1.15 \times 10^{12}$ GC ml$^{-1}$ for pENN.AAV.CaMKII.0.4.Cre.SV40 and AAVrh8-synP-DIO-sTpEpB-WPRE-bGH, respectively, and $1.7 \times 10^{10}$ GC ml$^{-1}$ for RVΔG-4mCherry (EnvA). One to two weeks (typically ~7 days) after the injection of the rabies virus, the animal was perfused, brain extracted, sectioned, immunostained, and imaged.

To validate projections from ACC to M2, we injected AAVretro-pmSyn1-EBFP-Cre (Addgene#51507-AAVrg) at M2 (1 µl, ML −1.3 mm, AP + 2.0 mm, DV −0.9 mm) and AAV5-hSyn-DIO-hM4Di-mCherry (Addgene#44362) at ACC (ML −0.5 mm, AP −1.0 mm, DV −1.4 mm). The viral titers were $1.1 \times 10^{13}$ GC ml$^{-1}$ for AAVretro-pmSyn1-EBFP-Cre and $4.7 \times 10^{12}$ GC ml$^{-1}$ for AAV5-hSyn-DIO-hM4Di-mCherry.

## Immunohistochemistry

Rats were deeply anesthetized using sodium pentobarbital and then transcardially perfused with saline and 4% paraformaldehyde (PFA). Brains were extracted and incubated in 4% PFA at room temperature overnight. Brains were transferred to PBS, and 50 µm coronal slices were prepared using a vibratome. For immunostaining, each slice was placed in PBS + 0.1% Triton X-100 (PBS-T), with 10% normal goat serum for 1 h and then incubated with primary antibody at 4 °C for 12 h. Slices then underwent three wash steps for 10 min each in PBS-T, followed by 2 h incubation with secondary antibody. After three more wash steps of 10 min each in PBS-T, slices were transferred to DAPI solution (5 µg ml$^{-1}$ in PBS), incubated for 30 min at room temperature, and then mounted on microscope slides. Antibodies used for staining were as follows: to stain for hM4Di-mCherry or mCherry alone, slices were incubated with primary rabbit anti-RFP (1:1000, Rockland) and visualized using anti-rabbit Alexa 555 or Alexa 568 (1:200). To stain for eNpHR3.0-eYFP and hChR2-eYFP, slices were incubated with primary chicken anti-GFP (1:1000, Life Technologies) and visualized using anti-rabbit Alexa 488 (1:200). Immuno-stained slices were imaged using an epifluorescence (Zeiss Imager.Z2) or a confocal microscope (Zeiss LSM700) with ×5 or ×10 objective lenses. Intensity of each

fluorescence channel in imaging data was adjusted using ZEN software (ZEN 2011 Blue edition, Zeiss).

## Chemogenetic experiments

Rats were bilaterally injected with AAV5-CaMKIIa-hM4Di-mCherry virus in ACC (area 24a'/24b') (Fig. 2a)[27,28]. At least 3 weeks after virus injection, animals were tested on CAS task with IP injections of either saline or CNO solutions. Ten or 20 mg kg$^{-1}$ solution of CNO (Sigma, C0832-5MG) was prepared by first dissolving CNO in dimethyl sulfoxide (DMSO; Sigma, 34869) followed by adding saline solution (final concentration of DMSO were 5% and 1% for IP injection and for cannula infusion experiments, respectively). CNO solutions were intraperitoneally injected 35–40 min before starting behavioral testing. In some sessions, we started behavior experiments 60 min after the IP injection of CNO solution to examine whether the duration from the CNO administration to the start time of behavioral testing could affect the animal's performance. In saline control conditions, a vehicle solution (5% DMSO in 0.9% saline) was intraperitoneally injected 35–40 min before behavioral sessions.

For locally infusing CNO solution or its control solution in M2 in rats infected with inhibitory DREADD virus in ACC, we implanted 26-gauge dual guide cannula (Plastics One) targeting bilateral M2 (ML ± 1.3 mm, AP + 2.0 mm) (Supplementary Fig. 9a, b). Rats were placed under a light non-surgical 1–1.5% isoflurane anesthesia and the CNO solution (0.5 µl, 1 µg µl$^{-1}$) was bilaterally infused through a 33-gauge dual internal cannula (Plastics One) at a speed of 0.2 µl min$^{-1}$. After waiting for 4 min after infusions, the infusion cannula was removed. Rats were kept in their home cage for 30 min until we started behavioral experiments.

For testing if an IP administration of CNO could suppress neural activity in ACC, we measured multiunit spiking activity in ACC in two rats infected with the inhibitory DREADD virus and implanted with silicon probes (Neuronexus) (Supplementary Fig. 2b). We first intraperitoneally injected a saline control solution (5% DMSO in 0.9% saline) immediately before starting neural activity measurements and continued the electrophysiological recording for 60 min. Then animals were transferred to the isoflurane chamber. After IP injection of the CNO solution (10 or 20 mg kg$^{-1}$ in 5% DMSO saline solution) under light anesthesia, we resumed neural activity measurements. Continuous voltage signals were recorded in the hard disc for offline data analysis. To obtain multiunit spike timestamps in offline analysis, the continuous voltage signals were high-pass filtered (400 Hz) digitally, and multiunit spikes were detected by thresholding the continuous signals at 4 SDs above the baseline level[46].

## Electrophysiology in task-behaving rats

The neural activity data were obtained from six rats in which Utah array electrodes with 4 × 8 matrix shanks (platinum or iridium-oxide tips; electrode length of 0.5 mm or 1mm; electrode spacing of 0.4 mm) (Blackrock Microsystems) were implanted in M2 targeting an area covering the Bregma coordinate of ML 1.3 mm, AP + 2.0 mm in either left or right hemispheres (left hemisphere in two animals and right hemisphere in three animals). This location was chosen because it was the center of the distribution of stimulation sites that resulted in contralateral orienting movement and neurons related to orienting responses were recorded in previous studies[8,47]. We also confirmed that delivering electrical stimulations (20 s$^{-1}$ bipolar injections of 30–60 µA current) at around this coordinate elicited contractions of the shoulders or limbs of the rats.

For implant surgery, animals were anesthetized in the isoflurane chamber and placed in the stereotaxic frame. After applying eye ointment and washing the incision sites with betadine and ethanol, an incision was created over the scalp and connective tissue was removed. The skull was drilled for attaching bone screws and for implanting the array electrodes in M2. After installing bone screws, we performed a

durotomy and slowly inserted the array electrodes at M2 using a manipulator and a mounting probe by applying a vacuum to steady the array. Dental cement was applied to secure the cable to the screws and the craniotomy was filled with a surgical silicone adhesive (Kwik-sil). The vacuum was turned off and the array was further secured using dental cement.

All recordings were conducted after the rats were fully recovered (at least seven days after surgery). The ground was taken from one of the skull screws typically above the cerebellum. The reference channel was chosen from one of the skull screws or one of the recording channels in which no clear spiking activity was observed. Data were recorded using a unity gain amplifier forwarded, filtered (600–8000 Hz, FIR filter), and stored in the Digital Lynx SX System (Neuralynx) for offline data analysis using Cheetah 5 software (Neuralynx).

## Optogenetic manipulation in behavior experiments
**(1) Optogenetic silencing of ACC neurons (Figs. 8 and 9).** Dual fiberoptic cannulae (dual 200 μm core, NA = 0.22, Doric Lenses) were implanted in the ACC in task-trained rats under isoflurane anesthesia (see surgery section for general procedures). A yellow-green laser (Opto Engine LLC, 561 nm) with a fiberoptic patch cable (dual 200 μm core, NA = 0.22, Doric Lenses) was installed in the behavioral chamber. A TTL pulse was delivered from the behavior system through an interface board (National Instrument) that determined the timing of laser light delivery for optogenetic intervention. Light was continuously delivered for 4 s during the outcome feedback period following the animals' 2nd choices (either incorrect 2nd choice or correct 2nd choice; see Fig. 8c, d). The output power of the laser to the bilateral fiberoptic cannula was calibrated to 15 mW per channel for the yellow-green laser with the implanted optical fiber attached. This power was determined by acute optogenetic experiments (see the following section: Unit recordings with optogenetic stimulation in acute condition).

**(2) Optogenetic excitation of ACC terminals in M2 cortex (Supplementary Figs. 14 and 16).** Dual fiberoptic cannulae (dual 200 μm core, NA = 0.22, Doric Lenses) were implanted in M2 cortex in task-trained rats under isoflurane anesthesia (see surgery section for general procedures). A blue laser (Opto Engine LLC, 473 nm) with a fiberoptic patch cable (dual 200 μm core, NA = 0.22, Doric Lenses) was installed in the behavioral chamber. A TTL pulse was delivered from the behavior system through an interface board (National Instrument) that determined the laser timing for optogenetic intervention. 473 nm light with 5 ms pulses was delivered at 10 or 20 Hz for 1 s during the pre-choice period (Supplementary Fig. 14) or for 4 s during the outcome feedback period following the animals' 2nd choices (Supplementary Fig. 16). The output power of the laser to the bilateral fiberoptic cannula was calibrated to 6 mW per channel for the 473 nm laser with the implanted optical fiber attached. This power was determined by acute optogenetic experiments (see the following section: Unit recordings with optogenetic stimulation in acute condition).

## Unit recordings with optogenetic stimulation in acute condition
To examine the effects of laser light delivery on neural activities in ACC in rats infected with eNpHR3.0-eYFP virus and hChR2-eYFP viruses, optrodes consisting of a tungsten electrode (0.5 or 1 MΩ; FHC Inc.) attached to an optical fiber (200 μm core diameter; Doric Lenses) with the tip of the fiber extending beyond the tip of the electrode by 200–300 μm were used for simultaneous optical stimulation and extracellular recordings in anesthetized condition. The optrode was slowly lowered to cingulate cortex where the eNpHR3.0 virus or hChR2 virus was injected. The optical fiber was connected to a yellow-green (561 nm) laser (for eNpHR3.0 virus) or to a blue (473 nm) laser (for

hChR2 virus) and was controlled by a beam shutter and a shutter controller. The power intensity of light emitted from the optrode was calibrated to 13–16 mW (for yellow-green laser) and 5–6 mW (for blue laser), respectively, as measured with an optical power/energy meter, which is consistent with the power intensity used in behavior experiments. Then, 561 nm light pulses were delivered at each depth of the optrode (pulses with 1 s duration at 5 Hz for eNpHR3.0 virus and with 5 ms duration at 10 or 20 Hz for hChR2 virus) while neuronal activity in ACC was collected for 10–20 sweeps. Continuous voltage data were monitored online using an oscilloscope and a sound speaker, fed into a preamplifier, transferred to an interface board, and saved in the hard disc. Continuous voltage traces were high-pass filtered with a Butterworth filter and then thresholded typically at around −50 to −80 μV using Offline Sorter software (Plexon). Spike rasters and peri-event time histograms (PETHs) of spiking activities of isolated single-units were plotted using Neuro Explorer software (version 4, Plexon). Rats were sacrificed, and brains were collected and sectioned for histological confirmation of recorded sites.

## Data analysis
Spike sorting was conducted using Offline Sorter software (Plexon). Spike rastergram (Fig. 8b) and some of the PETHs (Fig. 8b and Supplementary Fig. 2b) were created using Neuro Explorer software (ver. 4, Plexon). Intensity of each fluorescence channel in imaging data from immuno-stained brain sections was adjusted using ZEN software (ZEN 2011 Blue edition, Zeiss). Some figure components were created using BioRender software with a license. ANOVA was conducted using codes written in R (R Studio). All the other analyses were conducted using MATLAB (2013b and 2019a, Mathworks Inc).

## Quantification of animals' task performance (1st choice performance and 2nd choice performance)
**(1) Quantification of 1st choice responses in 1 step and 2 steps conditions.** Animals' task performance of the first choices in 1 step rule or 2 steps rule blocks (i.e., mapping auditory cue stimuli to 1st choice responses) was quantified using a percent error rate in their 1st choice responses (i.e., %1st choice error). Note that, in 1 step condition, animals were required to make a single choice response (i.e., go to the left- or right-side port) and thus there is no correct 2nd choice although, for convenience, we referred to the correct choice responses in 1 step condition as correct 1st choice responses; indeed, in 1 step condition, committing a 2nd response (i.e., a poking response to the side port opposite to the one chosen in the 1st choice) was not rewarded and, instead, an error feedback tone was provided. In group analysis, the percent 1st choice error rate for two trial types (i.e., trials with tone cue 1 and those with tone cue 2) were separately calculated and then averaged.

**(2) Quantification of 2nd choice responses in 2 steps condition (2nd choice omission error).** In addition, in 2 steps rule blocks, after an animal makes a correct 1st choice response, animals can make two kinds of choice responses. (1) Choose a side port opposite to the one that the animal chose in its 1st response (correct 2nd choice response). (2) Alternatively, animals may omit poking to the side port and, instead, may push the center lever for initiating the next trial (incorrect 2nd choice response). Such a response was classified as an incorrect 2nd choice (or a 2nd choice omission error) and was used as a behavioral measure to quantify the animal's ability to adapt to switches from 1 step to 2 steps conditions (see Supplementary Fig. 1d, right panel). We often observed such omission errors (i.e., incorrectly pushing the center lever without choosing the side port as a 2nd choice response) in trials immediately following switches from 1 step rule to 2 steps conditions (see Supplementary Fig. 3 for representative session data) because animals tend to continue making choice responses based on the old 1 step rule for some trials even after the rule has been

switched (note that a 2nd choice omission error in the 1st trial after the rule switch is inevitable, e.g., an inevitable error, because no explicit signal was provided to the animals for rule switches across blocks). In group analysis, like %1st choice error, %2nd choice omission error for two trial types (i.e., trials with tone cue 1 and those with tone cue 2) was separately calculated and then averaged.

**(3) Quantification of 2nd choice responses in 1 step condition (2nd choice commission error).** We also measured the frequency of animals making a 2nd choice response in 1 step rule block. Such response actions were rewarded in 2 steps condition, but they were not rewarded but instead were penalized by a presentation of feedback buzzer sound and by the prolonged ITI duration in 1 step condition. They were classified as 2nd choice commission error (see Supplementary Fig. 1d, left panel). We often observed such errors (i.e., poking the side port as a 2nd choice response) in trials immediately following switches from 2 steps rule to 1 step conditions because animals tended to continue making choice responses based on the old 2 steps rule even after the rule has already been switched (see Supplementary Fig. 3 for representative session data). We used the number of occurrences of 2nd choice commission error per trial as an operational measure to quantify an animal's capacity to adjust its choice responses following rule changes from 2 steps to 1 step conditions. This behavioral measure is expected to decrease in 1 step rule block as rats adjusted their responses after the rule switches from 2 steps to 1 step conditions. In group analysis, the count of 2nd choice commission errors per trial for two trial types (i.e., trials with tone cue 1 and those with tone cue 2) was separately calculated and then averaged.

### Quantification of response time for 1st and 2nd choices
Response time for the 1st choice response (RT1) was calculated as the difference in the timing of center lever entry and first-choice port entry (Choice 1). Similarly, response time for the 2nd choice response (RT2) was calculated as the difference in the timing of second choice port entry (Choice 2) and first-choice port entry (Choice 1).

### Inclusion/exclusion criteria of experimental sessions for group analysis
To compare animals' task performance between the 1st block in the session and the following Rule Switch-blocks (either 1 step or 2 steps rule block), sessions were not included in the group analysis if they do not have Rule Switch-blocks (i.e., if animals stopped task behaviors before the task entered in Rule Switch-blocks). Also, to compare animals' task performance across three epochs within a block (either 1 step or 2 steps rule -block), sessions were not included in the group analysis if they have less than 10 trials in the 3rd epoch of at least one Rule Switch-block. To compare the task performance in the first, middle, and third sections in a block, a block was subdivided into three epochs: trial no. 1–18 (1st), 19–36 (2nd), and 37–55 (3rd) for sessions with rule switches in every 55 trials. Similarly, in two sessions in which the rule was switched in every 40 trials, a block was subdivided into three epochs: trial no. 1–13 (1st), 14–26 (2nd), and 27–40 (3rd).

In some sessions, behavior experiments started from the 2 steps condition (1st block) and then switched to the 1 step condition (2nd block) and so forth (see Supplementary Fig. 1a). These sessions did not have 1st block (non-Rule Switch-block) for 1 step condition. Therefore, if all the sessions tested with an animal started with 2 steps rule block and did not have any session starting with 1 step condition, this animal was not included in the analysis of the 1st block (i.e., non-Rule Switch-block) for 1 step condition. Indeed, four animals out of nine were not included in this plot because, in these animals, all sessions started with 2 steps rule block and so the 1st block (i.e., non-Rule Switch-block) did not exist in 1 step condition. For the same reason, in Fig. 2c, there was no 1st block in 1 step condition because the sessions started with 2 steps condition, and so no thin blue/pink line. Similarly, some

experiments started from the 1 step condition (1st block) and then proceeded to the 2 steps condition (2nd block) and so forth. These sessions did not have 1st block (non-Rule Switch-block) for 2 steps condition. Therefore, if all the sessions tested with an animal started with 1 step rule block and did not have any session starting with 2 steps condition, this animal was not included in the analysis of the 1st block (i.e., non-Rule Switch-block) for 2 steps condition. Indeed, one animal out of nine (animal no. 6 in Supplementary Fig. 5a) was not included in this plot of Supplementary Fig. 5f because, in this animal, all sessions started with 1 step rule block and so 1st block (i.e., non-Rule Switch-block) did not exist in 2 steps condition.

### Chemogenetics behavior data analysis
We used a total of 31 rats for chemogenetics experiments; 15 rats were injected with the inhibitory DREADD virus (AAV5-CaMKIIa-hM4Di-mCherry) in ACC, 5 rats were injected with the mCherry control virus (AAV5-CaMKIIa-mCherry) in ACC, 6 and 5 rats were injected with the inhibitory DREADD virus in prelimbic/infralimbic cortex and ventral thalamic nuclei, respectively. Among the 15 rats that were injected with the inhibitory DREADD virus in ACC, 11 rats were used for experiments intraperitoneally administering CNO solutions ($n = 3$ rats for only 20 mg kg$^{-1}$ dosage, $n = 3$ rats for only 10 mg kg$^{-1}$ dosage, and $n = 5$ rats for both 20 and 10 mg kg$^{-1}$ dosage) while 5 rats were used for experiments locally infusing CNO solutions with a dose of 1 μg μl$^{-1}$ (one rat was used for both IP injection experiment and cannula infusion experiment). Two animals (three sessions) were also tested with an IP injection of CNO with a dose of 40 mg kg$^{-1}$ but the data were not included in the group analysis. Also, one rat was pilot-tested with a local infusion of CNO solutions with a dose of 0.4 and 4 μg μl$^{-1}$ but these data were not included in the group analysis.

For group analysis of the experiments with the inhibitory DREADD virus in ACC with IP injections of CNO (presented in Fig. 2d–k and Supplementary Fig. 5b–i), eight rats were tested with a CNO dose of 20 mg/kg (5 out of 8 rats were also tested with a CNO dose of 10 mg/kg). Another three rats were tested only with a CNO dose of 10 mg/kg. Only animals that were tested with CNO in at least two sessions were included in the group analysis (thus animal no. 1 and animal no. 10 in Supplementary Fig. 5a were not included). In the group analysis presented in Fig. 2d–k, sessions with CNO doses of 10 and 20 mg/kg were combined.

### Electrophysiology data analysis (chronic recordings from task-behaving rats)
**(i) Inclusion/exclusion criteria for trials.** In group analysis, as in chemogenetics behavior data analysis, sessions with a total trial number of less than 10 trials in the 3rd epoch of Rule Switch-block were not included in the analysis. Trials with outlier RT1 (cutoff = 3 s) were removed from electrophysiology data analysis (also see the following section. In all analyzed sessions, the proportion of trials with outlier RT1 was less than 2% of the total trials of the session).

**(ii) Spike sorting.** Single-units were isolated by spike sorting based on peak or valley and/or principal components of the voltage-thresholded waveforms using the Offline Sorter software (Plexon). Only a unit with a refractory period (>2 ms) in the auto-correlogram was accepted as single-units[46,48]. We analyzed neural data collected in 43 sessions from 5 rats: 29 sessions with saline control ($n = 5$ rats, 2–9 sessions for each), 14 sessions with an IP injection of CNO solution with a dose of 20 mg kg$^{-1}$ ($n = 5$ rats, 1–4 sessions for each). From these session data, a total of 900 single-units were isolated: 594 units and 306 units for saline and CNO conditions, respectively. All these units were included in the group analysis of mean firing rate (Fig. 5). Spike timestamps of a single-unit in each task trial were smoothed using a Gaussian kernel ($\sigma = 60$ ms) and PETHs were constructed with a bin width of 50 ms.

**(iii) Quantification of rule selectivity of single-units (Fig. 6 and Supplementary Figs. 12 and 13).** A single-unit was included in the group analysis of rule selectivity (Fig. 6) if it showed mean firing rate of at least 3 spikes s$^{-1}$ in the 1 s window immediately before the animals made the choice in either cue stimulus condition in 1 step condition or before animal's making the 1st choice in either cue stimulus condition in 2 steps condition (437 units and 195 units for saline and CNO conditions, respectively).

To quantify rule selectivity of neural responses, we used ROC analysis[29–31]. An ROC analysis measures the degree of overlap between two response distributions. For each M2 single-unit the preferred and non-preferred rule conditions were compared, given two distributions, P and N respectively, of neuronal activity. For example, for some neurons (e.g., a representative neuron presented in Fig. 5b, c), these distributions were the neurons' firing rates during the 2 steps rule in effect in comparison to the 1 step rule. An ROC curve was then generated by taking each observed firing rate of the neuron and plotting the proportion of P that exceeded the value of that observation against the proportion of N that exceeded the value of that observation. The area under the ROC curve was then calculated. A value of 0.5 would indicate that the two distributions were completely overlapped, and thus the neuron is not selective to the rules. A value of 1.0, on the other hand, would indicate that the two distributions were completely separated (i.e., every value drawn from N is exceeded by the entirety of P, whereas none of the values of P is exceeded by any of the values in N) and so the neuron is very selective. This method of analysis has the advantage that it is independent of the firing rate, and so can be used to compare neurons with different baseline firing rates and dynamic ranges[31]. It is also nonparametric and so does not require the distributions to be Gaussian.

We calculated the mean firing rate of each single-unit during a 1 s period immediately before animals made their 1st choices in each trial of either 1 step or 2 steps conditions (pre-choice period). We then calculated the area under ROC curve (auROC) using these firing rates. To examine how rule selectivity changes throughout the Rule Switch-blocks, a block was subdivided into three epochs: trial nos. 1–18 (1st epoch), 19–36 (2nd epoch), and 37–55 (3rd epoch) for sessions with rule switches in every 55 trials. Similarly, in two sessions in which the rule was switched in every 40 trials, a block was subdivided into three epochs: trial nos. 1–13 (1st epoch), 14–26 (2nd epoch), and 27–40 (3rd epoch).

Then ROC analysis was conducted for each epoch separately. For example, to calculate rule selectivity in the 1st epoch of a block following rule switches from 1→2 steps conditions, ROC curves were calculated using distributions of mean firing rates during the pre-choice period of trials in the 2nd and 3rd epochs in the preceding 1 step rule block and of mean firing rates during the pre-choice period of trials in the 1st epoch of subsequent 2 steps rule block. Similarly, rule selectivity in the 2nd (or 3rd) epoch of a block following rule switches from 1→2 steps conditions were calculated using distributions of mean firing rates during the pre-choice period of trials in the 2nd and 3rd epochs in the preceding 1 step rule block and of mean firing rates during pre-choice period of trials in the 2nd (or 3rd) epoch of subsequent 2 steps rule block. Similarly, rule selectivity in each of three epochs of a block following rule switches from 2→1 steps conditions was calculated using distributions of mean firing rates during the pre-choice period of trials in the 2nd and 3rd epochs in the preceding 2 steps rule block and of mean firing rates during the pre-choice period of trials in each of three epochs of subsequent 1 step rule block.

To obtain time courses of rule selectivity, we calculated mean firing rates of each neuron using a sliding window with 250 ms width and a step size of 50 ms throughout a trial and auROC curve was calculated for each time point[30]. To test the statistical significance of rule selectivity, we used a bootstrap analysis and repeated the ROC analysis 200 times to obtain 95% percentile threshold in which we assigned the rule condition (i.e., 1 step or 2 steps) at random to each trial and calculated the auROC. In group analysis, trial conditions were grouped in ipsilateral and contralateral -conditions depending on the hemisphere implanted with the electrode array in each animal: in ipsilateral trial conditions, animals were required to first choose the ipsilateral side port and then the contralateral side port as its 2nd choice (i.e., trials requiring the animal to select the right port as its 1st choice if the animal was implanted with the array electrode in the right hemisphere, and vice versa) while, in "contralateral" trial conditions, they were required to select the contralateral side port as its 1st choice and then to the ipsilateral side port as its 2nd choice (i.e., trials requiring the animal to go to the left port if the animal was implanted with the array electrode in the right hemisphere, and vice versa) (see Fig. 5a).

**(iv) Control of firing rates in quantifying rule selectivity of single-units (Supplementary Fig. 13).** To examine whether the observed differences in the rule selectivity as measured by the auROC merely reflect the difference in the firing rates of M2 neurons between CNO and saline conditions, we matched the distributions of firing rates of M2 neurons between CNO and saline conditions by randomly removing spikes measured in saline condition. This was conducted by the following procedures: (1) all the single-units included in the single-unit database were sorted in descending order according to their mean firing rates. (2) The firing rates of single-units measured in the same percentiles were paired between CNO and saline conditions. (3) Then, the spikes of single-units in saline condition were randomly removed so that the mean firing rates across trials were matched between the two single-units in the same percentiles of the sorted lists of neurons measured in CNO and saline conditions. (4) Once the firing rates were matched between CNO and saline conditions, an auROC was estimated in the same manner as in Fig. 6 (also see the following section: Quantification of rule selectivity of single-units). (5) We repeated the calculations in (3) and (4) 100 times and calculated a mean of 100 samples of ROC values for each neuron. This averaged value was used for comparing ROC between saline and CNO conditions in Supplementary Fig. 13.

**(v) Analysis of outcome-related neurons in M2 (Fig. 7 and Supplementary Fig. 15).** All the isolated M2 single-units were included in the group analysis (594 units and 306 units for saline and CNO conditions, respectively). They were classified according to their activity in the outcome feedback period. Each single-unit was tested if the mean firing rates in the outcome feedback period were significantly greater or smaller than those in the baseline period (paired $t$-test, one-sided, $P = 0.025$). Positive outcome-activated neurons were defined as single-units that showed significantly greater firing rates in the positive outcome feedback period (a 3 s period following a reward delivery after animals' making 2nd choices in 2 steps rule blocks) than those in the baseline period. Similarly, positive outcome-suppressed neurons were defined as single-units that showed significantly smaller firing rates in the positive outcome feedback period (a 3 s period following a reward delivery after animals' making 2nd choices in 2 steps rule blocks) than those in the baseline period. Negative outcome-activated neurons and negative outcome-suppressed neurons were defined in similar manners except that, for these neurons, the outcome feedback period was a 3 s period following an error feedback tone instead of a reward delivery (negative outcome feedback period). In Fig. 7c, mean firing rates in a 3 s period following error feedback tone were calculated for each negative outcome-activated neuron. Due to the limited number of incorrect 2nd choice trials in 2nd and 3rd epochs, trials in these epochs were combined in plotting population-averaged PETHs (Fig. 7b) and in comparing mean firing rates across epochs (Fig. 7c).

Note that two neurons among 21 negative outcome-activated neurons were not included in the analyses in Fig. 7b and the plots presented in Fig. 7b (bottom left panel) and in Fig. 7c (only 19 neurons

were included in the 1st epoch of CNO sessions) because there was no 2nd choice omission error trial in the 1st epoch in the corresponding session although these two neurons were classified as negative outcome-activated neurons and were included in the proportion results presented in Fig. 7a based on the activity in all three epochs in Rule Switch-blocks.

## Optogenetics data analysis

**(1) Optogenetic silencing of ACC neurons.** 561 nm laser light was delivered for 4 s upon animal's making an incorrect 2nd choice (Fig. 8c) or upon a correct 2nd choice (Fig. 8d). We quantified the percent error rate of 1st choice (%1st choice error) or 2nd choice (%2nd choice omission error) in trials immediately following trials in which a laser light was delivered upon animal's committing an incorrect 2nd choice (Fig. 9b, d) or making a correct 2nd choice (Fig. 9c, e). As in the analysis of chemogenetics data and electrophysiology data, for comparing the task performance in trials immediately following rule switches and trial in later part of the block, trials were categorized into three groups according to their positions in the block: trial nos. 1–18 (1st epoch), 19–36 (2nd epoch), and 37–55 (3rd epoch).

**(2) Optogenetic stimulation of ACC terminal in M2.** 473 nm laser light was delivered during the pre-choice period (i.e., 1 s period immediately after a presentation of tone cue stimulus) (Supplementary Fig. 14), upon animal's making an incorrect 2nd choice (Supplementary Fig. 16a, c, e) or upon a correct 2nd choice (Supplementary Fig. 16b, d, f). Quantifications of animals' performance were the same as in those for optogenetic silencing data (see the following section: Optogenetic silencing of ACC neurons). Sessions with light pulse frequencies of 10 and 20 Hz were combined in data analysis (in both frequency conditions, we delivered 473 nm light with a pulse width of 5 ms).

## Statistics

MATLAB (2013b, 2019a, Mathworks) and R (R Studio) were used for data analysis. No statistical analysis was conducted to pre-determine the sample sizes of each experiment. Statistics were run two-sided except mentioned otherwise. In testing the statistical significance of the effect of CNO dose or epoch in Rule Switch-blocks or their interaction in task performance (Figs. 2g–k and 4d–g), we conducted repeated measures two-way ANOVA with both CNO dose and epoch being within-subject factors by using aov function provided in R. In testing statistical significance of the effect of CNO dose or epoch in Rule Switch-blocks or their interaction in rule selectivity (Fig. 6e), we conducted a repeated measures two-way ANOVA with CNO dose and epoch being between-subject and within-subject factors, respectively, by using Anova function in the car library in R. All other statistical tests were conducted using MATLAB.

## Reporting summary

Further information on research design is available in the Nature Research Reporting Summary linked to this article.

## Data availability

Data used in this study are available at https://doi.org/10.17605/OSF.IO/EVC73. Source Data are provided with this paper.

## Code availability

Behavioral task program codes are available at https://doi.org/10.5281/zenodo.6618578. Data analysis codes are available at https://doi.org/10.17605/OSF.IO/EVC73.

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

## Acknowledgements

We thank J. Derwin, J. Martin, S.Y. Huang, and M. Ragion for help with experiments; K. Rockland and J. Yamamoto for experimental advice; M. Pignatelli, Q. Ferry, A. Aqrabawi, and K. Flick for comments on the manuscript and all the members of the Tonegawa lab for their support. This work was supported by the RIKEN Center for Brain Science, the Howard Hughes Medical Institute, the JPB Foundation (to S.T.), and the Human Frontier Science Program fellowship (to D.T.). Some figure items in this paper were created with BioRender.com.

## Author contributions

D.T., D.R., S.M., T.K., A.B., and S.T. designed the experiments. D.T. and C.L. conducted immunohistochemistry and image data acquisition. H.A.S. and I.R.W. constructed rabies virus. D.T. collected and analyzed behavior, electrophysiology, and circuit tracing data. D.T. and S.T. wrote the paper with input from all authors. All authors discussed and commented on the manuscript.

## Competing interests

The authors declare no competing interests.

## Additional information

**Peer review information** *Nature Communications* thanks anonymous, reviewer(s) for their contribution to the peer review of their work. Peer review reports are available.

