## [Peer Review File · Nature Communications]

REVIEWERS' COMMENTS

Reviewer #1 (Remarks to the Author):

Thanks for addressing my concerns.

Reviewer #2 (Remarks to the Author):

The authors addressed most of my major comments from the previous revision, and I don't have major outstanding technical concerns. They adjusted the language so that it is more readable and accessible (e.g., replacing %Error2 and Nr2a with errors of omission/comission). There are some interesting observations in this study, for instance, that ACC is required when transitioning from simpler to more complicated rules or action sequences, but not the other way. I think the fact that the rule encoding in M2 is only observed for ipsilateral choices is very interesting, albeit under-explored in this manuscript, and it's a cool result that the encoding is diminished if ACC is inactivated. However, many of the results are difficult to reconcile with one another, which makes it difficult to draw strong interpretable conclusions from the study. This is my major outstanding concern, which I'll try to articulate below.

The analysis of the electrophysiology data is very difficult to interpret. For instance, on page 12, line 6: "Taken together, these results showed that chemogenetic silencing of ACC increased activity of negative outcome-activated M2 neurons specifically during negative outcome feedback period following animals' incorrect 2nd choices upon rule switches from 1 step to 2 steps conditions." It is not clear what this means for behavior. Does this result connect to behavior at all? If anything, I would expect that increasing the activity of negative outcome-activated neurons following a rule-switch would increase behavioral flexibility, but they find the opposite with CNO.

Moreover, I don't understand how the previous finding relates to the observation that "the proportion of positive outcome-suppressed neurons was greater in CNO condition than in saline condition." It seems like after doing a number of pairwise comparisons, they find that there are more positive outcome-suppressed neurons with CNO, although I don't know if/how that relates to their behavioral effect. If they continue doing pairwise comparisons on smaller subsets of trials, they find that negative outcome-activated neurons exhibit enhanced activity during negative outcome period after rule switches with CNO. Again, I don't know if/how this relates to the behavioral effect.

Finally, on page 13, line 11: "Contrary to the 12 results from optogenetic silencing of ACC neurons (Fig.9), optogenetic excitations of ACC-M2 projections using ChR2 during outcome feedback period did not affect the 2nd choice performance in the immediately following trials (Extended Data Fig. 16c,d)." How are we to interpret this result?? It is at odds with the halorhodopsin and DREADDs data.

It is not the job of the reader to connect the dots between disparate pieces of complicated data. The fact that the results are often difficult to parse and reconcile with one another limits the interpretability, impact, and broad appeal of the work. If the authors can manage to tie the interesting pieces together more strongly, particularly connecting the electrophysiology to the inactivation experiments and behavior, I think there are enough interesting insights to warrant publication in Nature Communications. But as it is written, it is challenging to interpret and appreciate the results, and I honestly don't know what I am supposed to take away from the electrophysiology data in

particular.

Minor comments:

Page 6 line 5: "(1-6th, 6-12th, 13-18th trials)"... should be 7-12th, no?

Figure 2: the difference between the block "epochs" and the subsets of trials was a little confusing. It might be helpful to the reader to include a schematic that describes how these relate to each other.

Figure 4e,f: it seems strange for a difference to emerge in the second epoch of the 2-steps block, but not in the first (4e) and in trials 13-18 in the 2-steps block, but not before. Do the authors have an interpretation of this result? If not, then what is added by including it? It seems like the data is quite noisy and maybe they need to do more experiments? I don't know what these panels are adding.

Reviewer #3 (Remarks to the Author):

The authors have satisfactorily addressed my concerns from the previous version. The manuscript is acceptable for publication in Nature Communications.

Point-by-point responses to the Reviewer #2

Reviewer #2 (Remarks to the Author):

“The authors addressed most of my major comments from the previous revision, and I don’t have major outstanding technical concerns. They adjusted the language so that it is more readable and accessible (e.g., replacing %Error2 and Nr2a with errors of omission/comission). There are some interesting observations in this study, for instance, that ACC is required when transitioning from simpler to more complicated rules or action sequences, but not the other way. I think the fact that the rule encoding in M2 is only observed for ipsilateral choices is very interesting, albeit under-explored in this manuscript, and it’s a cool result that the encoding is diminished if ACC is inactivated.”

Response

We are glad that we could address most of the reviewer’s major concerns from the previous revision. Also, we appreciate the reviewer’s interests and positive appraisals in some of the observations presented in this study.

Major concerns:

“However, many of the results are difficult to reconcile with one another, which makes it difficult to draw strong interpretable conclusions from the study. This is my major outstanding concern, which I’ll try to articulate below.

The analysis of the electrophysiology data is very difficult to interpret. For instance, on page 12, line 6: “Taken together, these results showed that chemogenetic silencing of ACC increased activity of negative outcome-activated M2 neurons specifically during negative outcome feedback period following animals’ incorrect 2nd choices upon rule switches from 1 step to 2 steps conditions.” It is not clear what this means for behavior. Does this result connect to behavior at all? If anything, I would expect that increasing the activity of negative outcome-activated neurons following a rule-switch would increase behavioral flexibility, but they find the opposite with CNO.”

Response:

We admit that the previous manuscript lacked a clear description regarding how the electrophysiology data could relate to the behavior data. In accordance with the Reviewer’s comment, we have added a description in the Result section regarding how the chemogenetic effect on negative outcome-activated neurons shown in Fig. 7 could relate to the previous behavioral results (page 12, lines 16-21). Also, as we elaborated in our responses below, we have added a discussion paragraph in the Discussion section (page 15, line 20-page 17, line 13) and have also added a figure (Figure 10) in which we depicted how neural activity data could relate to behavior data.

“Moreover, I don’t understand how the previous finding relates to the observation that “the proportion of positive outcome-suppressed neurons was greater in CNO condition than in saline condition.” It seems like after doing a number of pairwise comparisons, they find that there are more positive outcome-suppressed neurons with CNO, although I don’t know if/how that relates to their behavioral effect.”

Response:

First, during the revision of the manuscript, we found that there was an error in the analysis of proportion statistics and its associated figure panels (Fig. 7a) and so have corrected them. In the previous analysis of proportion statistics, we used a single-unit database in which only neurons that showed a baseline firing rate greater than a threshold (1 Hz) were included. On the other hand, all the other analyses presented in Fig. 7b and c, we used a single-unit database in which all the recorded single-units were included. So, in the previous manuscript, the single-unit database was not consistent between the analyses presented in Fig. 7a and 7b,c. This mistake occurred because we have conducted sensitivity analysis on the threshold parameter for baseline firing rate and there were several versions of the results (we confirmed that the results were qualitatively very stable to the choices of threshold parameter). In this revision, we have replaced the proportion statistics shown in Fig. 7a with the one in which the same single-unit database was used as in the analysis of Fig. 7b,c. We sincerely apologize for this mistake.

Overall, the results are quite like the previous ones in their overall tendency although there were changes in the descriptions of statistical significance: The proportion of positive outcome-activated neurons was smaller in CNO condition than in saline condition (20.0% vs 13.1%, $\chi^2 = 6.73$, $P = 0.009$). The proportion of positive outcome-suppressed neurons was greater in CNO condition than in saline condition (35.5% vs 48.4%, $\chi^2 = 13.9$, $P = 0.00019$). The proportion of negative outcome-activated neurons was smaller in CNO condition than in saline condition (11.1% vs 6.9%, $\chi^2 = 4.17$, $P = 0.041$). The proportion of negative outcome-suppressed neurons was smaller in CNO condition than in saline condition but was not statistically significant (15.7% vs 19.9%, $\chi^2 = 2.61$, $P = 0.106$). These results showed that a chemogenetic silencing of ACC neurons decreased the proportion of outcome-activated neurons and increased that of outcome-suppressed neurons irrespective of whether the outcome feedback was positive or negative.

Then, how do these proportion data could relate to previous behavioral data? We think that the reduction of proportions of outcome-activated M2 neurons may affect an animal’s capacity to use outcome information (either positive or negative) for rule updating in some ways and this could affect animals’ 2nd choice performance. Importantly, the epoch-specific effect of ACC silencing was observed only in negative outcome-activated neurons and so other outcome encoding neurons cannot explain the epoch-specific disruption of animals’ 2nd choice performance in a simple manner. Given such difference, it might be that a decrease in the proportion of positive outcome-activated M2 neurons may attenuate an animal’s capacity to use positive outcome information for maintaining the representation of the currently working rule throughout the block. On the other hand, a decrease in the proportion of negative outcome-activated M2 neurons might attenuate an animal’s capacity to use negative outcome information for updating rule representation upon rule switches.

Please note that we have addressed the same issue (i.e., outcome activity in M2 neurons and behavior data) in our responses to the reviewer's next major comment because we needed to consider not only the proportion data (Fig. 7a) but also the dynamics data (Fig. 7b,c) to address how the negative outcome-activated M2 neurons could affect animals' behavior in epoch-specific manner.

We have revised the text in which we modified the description of the results of proportion statistics (page 11, lines 11-21). We have also added the discussion that we described above in the Discussion section (page 15, line 20- page 17, line 13).

"If they continue doing pairwise comparisons on smaller subsets of trials, they find that negative outcome-activated neurons exhibit enhanced activity during negative outcome period after rule switches with CNO. Again, I don't know if/how this relates to the behavioral effect."

Response:

Thank you for the feedback. We have added a discussion paragraph in the Discussion section in which we have elaborated possible mechanisms regarding how the enhanced activity of negative outcome-activated neurons could relate to the behavioral effect (page 15, line20-page 17, line 13). To support this discussion, we used the newly added Figure 10 (for details about this new figure, please see our response to the reviewer's last major comment below). Briefly, by using the concepts of evidence accumulation models, we argued that (1) the enhanced activity of negative outcome-activated neurons might be associated with an increase of threshold for rule updating (Fig. 10d) and that (2) the epoch-specific disruption of animals' 2nd choice performance induced by ACC silencing (Figs. 2, 4 and 9) might reflect the excessive increase of activity of negative outcome-activated neurons (Fig. 7b,c). This argument is based on the observation that the enhancement of activity of negative outcome-activated neurons was specifically observed in the 1st epoch of rule switch blocks and thus is consistent with the behavioral data while such epoch-specific effect was not observed in other outcome encoding neurons.

"Finally, on page 13, line 11: "Contrary to the 12 results from optogenetic silencing of ACC neurons (Fig.9), optogenetic excitations of ACC-M2 projections using ChR2 during outcome feedback period did not affect the 2nd choice performance in the immediately following trials (Extended Data Fig. 16c,d)." How are we to interpret this result?? It is at odds with the halorhodopsin and DREADDs data."

Response

Thank you for the feedback. The observation that an optogenetic suppression of ACC neurons but not an excitation of ACC-M2 projections affected the 2nd choice performance in trials immediately following rule switches suggests several possibilities.

An optogenetic excitation of ACC-M2 projections during the outcome period did not affect animals' choice performance while an optogenetic suppression of ACC neurons during the same task

period could affect it (Fig. 9 and Fig. S16). This discrepancy raises several possibilities. First, the expression of ChR2 in ACC neuronal axons terminating in M2 may not have been strong enough to elicit behavioral effect. Second, the spread of excitation light at M2 may not have been broad enough and thus could not excite large portions of ChR2-expressing ACC axon terminals. However, considering that the same stimulation protocol could affect the task performance when the light was delivered during the pre-choice instead of the outcome feedback period (Fig. S15), these two possibilities are not very likely. Instead, it seems more likely that the ChR2 stimulation protocol used in this study was not optimal. The observation that a chemogenetic silencing of ACC neurons did not decrease but increased the activity of negative outcome activated M2 neurons (Fig. 7b,c) suggest that ACC may modulate these neurons via disinhibition circuits (Fig. 10b) (note that, among all four types of outcome encoding M2 neurons, only negative outcome-activated neurons showed a difference in the firing rate between CNO and saline conditions in epoch-dependent manner; see Fig. 7b,c). If this is the case, a ChR2 stimulation at ACC terminals in M2 is expected to affect animals' performance only if it drives sufficiently large portions of inhibitory M2 neurons that receive ACC afferents. While we used a relatively short pulse duration in ChR2 excitation (i.e., 5 ms) which has been shown to be effective for exciting fast-spiking interneurons, the pulse frequency (10 or 20 Hz) may have been suboptimal for driving these neurons (Ref. Sohal et.al, 2009). Therefore, although we did not see a behavioral effect of optogetic excitations of ACC-M2 projections during the outcome period, a similar experiment with a higher pulse frequency may be able to show an effect on animals' task performance. Including this issue, precise circuit mechanisms regarding how ACC-M2 circuits work in rule switching remains unexplored and it is an open question whether and how ACC neurons could modulate downstream M2 neurons possibly in cooperation with local inhibitory neuronal networks.

We have added these discussion in the Discussion section (page 20, line 10-page 21, line 9).

Reference

Sohal, V. S., Zhang, F., Yizhar, O. & Deisseroth, K. Parvalbumin neurons and gamma rhythms enhance cortical circuit performance. *Nature* 459, 698-702, doi:10.1038/nature07991 (2009).

“It is not the job of the reader to connect the dots between disparate pieces of complicated data. The fact that the results are often difficult to parse and reconcile with one another limits the interpretability, impact, and broad appeal of the work. If the authors can manage to tie the interesting pieces together more strongly, particularly connecting the electrophysiology to the inactivation experiments and behavior, I think there are enough interesting insights to warrant publication in Nature Communications. But as it is written, it is challenging to interpret and appreciate the results, and I honestly don't know what I am supposed to take away from the electrophysiology data in particular.”

Response:

We appreciate for the reviewer's candid opinion. We have added a new Figure (Figure 10) as attached below in which we tried to more strongly connect the electrophysiology data to the inactivation experiments so that readers can easily grasp the essential points of this study. In this new figure, we illustrated two possible mechanisms regarding how a silencing of ACC neurons could affect animals' task switching performance by modulating outcome encoding M2 neurons.

Possible mechanisms of how a silencing of ACC neurons affects animals' task switching performance by modulating rule representations in M2 circuit. **a**, A possible ACC-M2 circuit in which a silencing of ACC neurons decreases activity of outcome encoding M2 neurons (positive outcome-activated neurons, positive outcome-suppressed neurons and negative outcome-suppressed neurons; see Fig. 7b for their activity profiles). Red down arrow indicates a decrease of activity. **b**, A possible ACC-M2 circuit in which a silencing of ACC neurons enhances activity of negative outcome-activated M2 neurons (see Fig. 7b,c for its activity profile). In this circuit model, when ACC neurons are silenced, a disinhibition mechanism in M2 increases the activity of negative outcome-activated M2. Red upward or downward arrow indicates an increase or a decrease of activity, respectively. **c**, An evidence accumulation model describing how a silencing of ACC neurons can affect an animal's rule switch performance. In this model, an animal updates the choice rule (i.e., 1 step or 2 steps rule) when the evidence for the new rule crosses a certain threshold level. The evidence for the new rule can increase stepwise every time the animal receives an outcome feedback after the animal makes 2nd choices. When ACC neurons are not silenced, after the animal experiences several error trials upon rule switches, the evidence for the new rule can cross the threshold, enabling the animal to make a rule switch (blue). In contrast, when ACC neurons are silenced, the step size of evidence accumulation for the new rule decreases, thus requiring more evidence accumulation (i.e., outcome feedback) before it crosses the threshold (red). Down arrows represent trial positions at which the evidence for the new rule crosses the threshold (blue, saline; red, CNO). **d**, Format is the same as in **c**, but, in this model, instead of decreasing the step size of evidence accumulation, a silencing of ACC neurons pushes up the threshold for rule updating, thus requiring more evidence accumulation for threshold crossing. Blue and red dotted lines, threshold for rule updating in saline and CNO conditions, respectively.

Minor comments:

Page 6 line 5: “(1-6th, 6-12th, 13-18th trials)”... should be 7-12th, no?

Response: Thank you for the correction. We have corrected the number.

Figure 2: the difference between the block “epochs” and the subsets of trials was a little confusing. It might be helpful to the reader to include a schematic that describes how these relate to each other.

Response:

Thank you for the constructive feedback. Following reviewer’s suggestion, we have added a schematic (on the right side of panel Fig. 2c; please see the panels as attached below) in which we described how the block epochs and the subset of trials relate to each other.

Figure 4e,f: it seems strange for a difference to emerge in the second epoch of the 2-steps block, but not in the first (4e) and in trials 13-18 in the 2-steps block, but not before. Do the authors have an interpretation of this result? If not, then what is added by including it? It seems like the data is quite noisy and maybe they need to do more experiments? I don't know what these panels are adding.

Response:

Thank you for raising these points. First, we think that the reason why the difference did not show up in the first epoch (Fig. 4e) was the low temporal resolution for block trials. Indeed, with higher temporal resolution (i.e., at a resolution of 6 trials-window instead of 18 trials-window), we did find the difference in the first epoch (Fig. 4f). Second, we think that the reason why the difference showed up in trials 13-18 in the 2-steps block but not before (Fig. 4e) was that it took 5-10 trials for animals to completely abolish the old rule and gets stabilized to the new rule even when ACC projections to M2 are intact (not silenced). We provide our thought on this issue below:

Ideally, animals should be able to make rule switching in the next trial after the animal makes an inevitable error due to the rule switch. However, in reality, it seems to take a while (around 5-10 trials on average) for an animal to completely abolish the old rule and switch to the new one even when ACC projections to M2 are intact (not silenced). This seems to be the reason why we did not see

the significant difference between saline and CNO conditions in animals' 2nd choice performance in earlier part of the first epoch.

We have added a discussion paragraph in the Discussion section in which we described our explanations on this issue (page 18, line 7- page 19, line 2).

REVIEWERS' COMMENTS

Reviewer #1 (Remarks to the Author):

Thanks for addressing my concerns.

Reviewer #2 (Remarks to the Author):

The authors addressed most of my major comments from the previous revision, and I don't have major outstanding technical concerns. They adjusted the language so that it is more readable and accessible (e.g., replacing %Error2 and Nr2a with errors of omission/comission). There are some interesting observations in this study, for instance, that ACC is required when transitioning from simpler to more complicated rules or action sequences, but not the other way. I think the fact that the rule encoding in M2 is only observed for ipsilateral choices is very interesting, albeit under-explored in this manuscript, and it's a cool result that the encoding is diminished if ACC is inactivated. However, many of the results are difficult to reconcile with one another, which makes it difficult to draw strong interpretable conclusions from the study. This is my major outstanding concern, which I'll try to articulate below.

The analysis of the electrophysiology data is very difficult to interpret. For instance, on page 12, line 6: "Taken together, these results showed that chemogenetic silencing of ACC increased activity of negative outcome-activated M2 neurons specifically during negative outcome feedback period following animals' incorrect 2nd choices upon rule switches from 1 step to 2 steps conditions." It is not clear what this means for behavior. Does this result connect to behavior at all? If anything, I would expect that increasing the activity of negative outcome-activated neurons following a rule-switch would increase behavioral flexibility, but they find the opposite with CNO.

Moreover, I don't understand how the previous finding relates to the observation that "the proportion of positive outcome-suppressed neurons was greater in CNO condition than in saline condition." It seems like after doing a number of pairwise comparisons, they find that there are more positive outcome-suppressed neurons with CNO, although I don't know if/how that relates to their behavioral effect. If they continue doing pairwise comparisons on smaller subsets of trials, they find that negative outcome-activated neurons exhibit enhanced activity during negative outcome period after rule switches with CNO. Again, I don't know if/how this relates to the behavioral effect.

Finally, on page 13, line 11: "Contrary to the 12 results from optogenetic silencing of ACC neurons (Fig.9), optogenetic excitations of ACC-M2 projections using ChR2 during outcome feedback period did not affect the 2nd choice performance in the immediately following trials (Extended Data Fig. 16c,d)." How are we to interpret this result?? It is at odds with the halorhodopsin and DREADDs data.

It is not the job of the reader to connect the dots between disparate pieces of complicated data. The fact that the results are often difficult to parse and reconcile with one another limits the interpretability, impact, and broad appeal of the work. If the authors can manage to tie the interesting pieces together more strongly, particularly connecting the electrophysiology to the inactivation experiments and behavior, I think there are enough interesting insights to warrant publication in Nature Communications. But as it is written, it is challenging to interpret and appreciate the results, and I honestly don't know what I am supposed to take away from the electrophysiology data in particular.

Minor comments:

Page 6 line 5: "(1-6th, 6-12th, 13-18th trials)"... should be 7-12th, no?

Figure 2: the difference between the block "epochs" and the subsets of trials was a little confusing. It might be helpful to the reader to include a schematic that describes how these relate to each other.

Figure 4e,f: it seems strange for a difference to emerge in the second epoch of the 2-steps block, but not in the first (4e) and in trials 13-18 in the 2-steps block, but not before. Do the authors have an interpretation of this result? If not, then what is added by including it? It seems like the data is quite noisy and maybe they need to do more experiments? I don't know what these panels are adding.

Reviewer #3 (Remarks to the Author):

The authors have satisfactorily addressed my concerns from the previous version. The manuscript is acceptable for publication in Nature Communications.